# Stochastic Deep Restoration Priors for Imaging Inverse Problems

Yuyang Hu [1]   Albert Peng [1]   Weijie Gan [1]   Peyman Milanfar [2]   Mauricio Delbracio [2]   Ulugbek S. Kamilov [1]

## Abstract

Deep neural networks trained as image denoisers are widely used as priors for solving imaging inverse problems. We introduce *Stochastic deep Restoration Priors (ShaRP)*, a novel framework that stochastically leverages an ensemble of deep restoration models beyond denoisers to regularize inverse problems. By using generalized restoration models trained on a broad range of degradations beyond simple Gaussian noise, ShaRP effectively addresses structured artifacts and enables self-supervised training without fully sampled data. We prove that ShaRP minimizes an objective function involving a regularizer derived from the score functions of minimum mean square error (MMSE) restoration operators. We also provide theoretical guarantees for learning restoration operators from incomplete measurements. ShaRP achieves state-of-the-art performance on tasks such as magnetic resonance imaging reconstruction and single-image super-resolution, surpassing both denoiser- and diffusion-model-based methods without requiring retraining.

## 1. Introduction

Many problems in computational imaging, biomedical imaging, and computer vision can be viewed as *inverse problems*, where the goal is to recover an unknown image from its noisy and incomplete measurements. Inverse problems are typically ill-posed, thus requiring additional prior information for accurate image reconstruction. While many approaches have been proposed for implementing image priors, the current research focuses on methods based on deep learning (DL) (McCann et al., 2017; Ongie et al., 2020; Kamilov et al., 2023; Wen et al., 2023).

Deep neural networks trained as image denoisers are widely-used for specifying image priors for solving *general* inverse

---

problems (Romano et al., 2017; Kadkhodaie & Simoncelli, 2021; Zhang et al., 2022). The combination of pre-trained Gaussian denoisers with measurement models has been shown to be effective in many inverse problems, including image super-resolution, deblurring, and medical imaging (see the recent reviews (Ahmad et al., 2020; Kamilov et al., 2023; Milanfar & Delbracio, 2024; Daras et al., 2024)). This success has led to active research on novel methods based on denoiser priors, their theoretical analyses, statistical interpretations, as well as connections to related approaches such as score matching and diffusion models (Venkatakrishnan et al., 2013; Chan et al., 2017; Romano et al., 2017; Buzzard et al., 2018; Reehorst & Schniter, 2019; Sun et al., 2019; Sun et al., 2019; Ryu et al., 2019; Cohen et al., 2021; Hurault et al., 2022b; Laumont et al., 2022; Gan et al., 2023a; Renaud et al., 2024; Xiao et al., 2024; Nathan et al., 2024).

Although priors based on Gaussian denoising models have been extensively studied, there is little research on leveraging priors from pre-trained restoration models that extend beyond Gaussian denoisers. In this paper, we present evidence that priors derived from deep models pre-trained as general restoration operators can surpass those trained exclusively for Gaussian denoising. We introduce a novel framework called *Stochastic deep Restoration Priors (ShaRP)*, which provides a principled approach to integrate an ensemble of general restoration models as priors to regularize inverse problems. By using more versatile restoration models, ShaRP improves upon traditional methods using Gaussian denoiser priors in two key ways: (a) ShaRP improved performance by using restoration models that are better suited for mitigating non-Gaussian structured artifacts arising during inference. (b) The restoration models in ShaRP can sometimes be directly trained in a self-supervised manner without fully-sampled measurement data.

We present novel theoretical and numerical results highlighting the potential of using an ensemble of restoration models as image priors. Our theoretical result introduces a novel notion of regularization for inverse problems corresponding to the average of likelihoods associated with the degraded observations of an image. The proposed regularizer has an intuitive interpretation as promoting solutions whose *multiple* degraded observations resemble realistic degraded images. We show that ShaRP seeks to minimize an objective function containing this regularizer. Our second theoreti-

---

[1]WashU [2]Google. Correspondence to: Ulugbek S. Kamilov <kamilov@wustl.edu>.

*Proceedings of the 42nd International Conference on Machine Learning*, Vancouver, Canada. PMLR 267, 2025. Copyright 2025 by the author(s).

cal result analyzes the possibility of learning the minimum mean squared error (MMSE) restoration operators directly from noisy and undersampled measurements. We numerically show the practical relevance of ShaRP by applying it to MRI reconstruction with varying undersampling patterns and rates, using a fixed-rate pre-trained MRI reconstruction network as a prior. We also show that ShaRP can use a pre-trained image deblurring model to perform single image super-resolution (SISR). Our numerical experiments show that ShaRP adapts the pre-trained restoration model as a prior, outperforming existing methods based on image denoisers and diffusion models, and achieving state-of-the-art results. Our experiments additionally highlight the benefit of using restoration models as priors by considering a setting where only undersampled and noisy MRI data is available for pre-training the prior. In such cases, self-supervised training of a restoration model is feasible, whereas training a Gaussian denoiser requires fully sampled data.

## 2. Background

**Inverse Problems.** Many computational imaging tasks can be formulated as inverse problems, where the goal is to reconstruct an unknown image $\boldsymbol{x} \in \mathbb{R}^n$ from its corrupted measurement

$$\boldsymbol{y} = \boldsymbol{A}\boldsymbol{x} + \boldsymbol{e}, \tag{1}$$

where $\boldsymbol{A} \in \mathbb{R}^{m \times n}$ is a measurement operator and $\boldsymbol{e} \in \mathbb{R}^m$ is the noise. A common approach to addressing inverse problems is to formulate them as an optimization problem

$$\widehat{\boldsymbol{x}} \in \underset{\boldsymbol{x} \in \mathbb{R}^n}{\arg\min} f(\boldsymbol{x}) \quad \text{with} \quad f(\boldsymbol{x}) = g(\boldsymbol{x}) + h(\boldsymbol{x}), \tag{2}$$

where $g$ is the data-fidelity term that quantifies the fit to the measurement $\boldsymbol{y}$ and $h$ is a regularizer that incorporates prior information on $\boldsymbol{x}$. For instance, typical functions used in imaging inverse problems are the least-squares term $g(\boldsymbol{x}) = \frac{1}{2} \|\boldsymbol{A}\boldsymbol{x} - \boldsymbol{y}\|_2^2$ and the total variation (TV) regularizer $h(\boldsymbol{x}) = \tau \|\boldsymbol{D}\boldsymbol{x}\|_1$, where $\boldsymbol{D}$ is the image gradient and $\tau > 0$ is a regularization parameter.

**Deep Learning.** DL has emerged as a powerful tool for addressing inverse problems (McCann et al., 2017; Ongie et al., 2020; Wen et al., 2023). Instead of explicitly defining a regularizer, DL methods use deep neural networks (DNNs) to map the measurements to the desired images (Wang et al., 2016; Jin et al., 2017; Kang et al., 2017; Chen et al., 2017; Delbracio et al., 2021; Delbracio & Milanfar, 2023). Model-based DL (MBDL) is a widely-used sub-family of DL algorithms that integrate physical measurement models with priors specified using CNNs (see reviews by (Ongie et al., 2020; Monga et al., 2021)). The literature of MBDL is vast, but some well-known examples include plug-and-play priors (PnP), regularization by denoising (RED), deep unfolding (DU), compressed sensing using generative models

(CSGM), and deep equilibrium models (DEQ) (Bora et al., 2017; Romano et al., 2017; Zhang & Ghanem, 2018; Hauptmann et al., 2018; Gilton et al., 2021a; Liu et al., 2022; Hu et al., 2024d). These approaches come with different trade-offs in terms of imaging performance, computational and memory complexity, flexibility, need for supervision, and theoretical understanding.

**Denoisers as Priors.** Score-based models (SBMs) are a powerful subset of DL methods for solving inverse problems that use deep Gaussian denoisers as imaging priors. Plug-and-Play (PnP) methods can be viewed as SBMs that incorporate denoisers within iterative optimization algorithms (see recent reviews (Ahmad et al., 2020; Kamilov et al., 2023)). These approaches construct a cost function by combining an explicit likelihood with a score function implicitly defined by the denoiser prior. Over the past few years, numerous variants of PnP have been developed (Venkatakrishnan et al., 2013; Romano et al., 2017; Metzler et al., 2018; Dong et al., 2019; Zhang et al., 2019; Wei et al., 2020; Hurault et al., 2022a), which has motivated an extensive research into their theoretical properties and empirical effectiveness (Chan et al., 2017; Buzzard et al., 2018; Ryu et al., 2019; Sun et al., 2019; Tirer & Giryes, 2019; Teodoro et al., 2019; Sun et al., 2021; Cohen et al., 2021; Fang et al., 2024; Renaud et al., 2024; Hu et al., 2024a; Terris et al., 2024). Diffusion Models (DMs) represent another category of SBMs; they are trained to learn the score function of the underlying probability distribution governed by stochastic differential equations (SDEs) (Ho et al., 2020; Song et al., 2021). Once trained, these models can be used as powerful priors for inverse problems by leveraging their learned score functions. Specifically, pre-trained DMs facilitate posterior sampling by guiding the denoising process to generate data consistent with observed measurements. This approach enables DMs to address inverse problems, often achieving impressive perceptual performance even for highly ill-posed inverse problems (Chung et al., 2023; Zhu et al., 2023; Wang et al., 2023; Feng et al., 2023; Sun et al., 2024; Wu et al., 2024; Song et al., 2024; Hu et al., 2024b; Alçalar & Akçakaya, 2024; Zhao et al., 2024; Rout et al., 2024; Bai et al., 2024; Mardani et al., 2024).

**Restoration Networks as Priors.** In addition to denoiser-based methods, recent work has also considered using restoration models as implicit priors for solving inverse problems (Zhang et al., 2019; Liu et al., 2020; Gilton et al., 2021b; Hu et al., 2024c). It has been observed that pre-trained restoration models can be effective priors for addressing unseen inverse problems, sometimes surpassing traditional denoiser-based approaches (Hu et al., 2024c). However, existing methods present two main limitations. First, existing restoration priors have relied on a fixed prior tailored to a specific degradation. Although deep restoration models can be trained in various settings—such as different

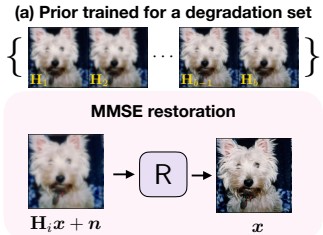
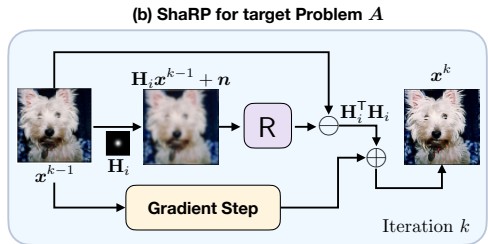

Figure 1: A restoration network trained on a set of tasks $\{\mathbf{H}_i\}$ can be used as a prior within ShaRP to address different target inverse problems without the need for retraining.

blur kernels for image deblurring or diverse undersampling masks for MRI reconstruction—current approaches do not leverage this capability, limiting their robustness to diverse artifacts. Second, prior work has not explored the potential of learning restoration priors directly from undersampled measurements, without access to fully sampled data. Unlike Gaussian denoisers, training without fully sampled data is natural for restoration models (Ulyanov et al., 2018; Krull et al., 2019; Quan et al., 2020; Yaman et al., 2020; Liu et al., 2020; Tachella et al., 2022; Chen et al., 2022; Millard & Chiew, 2023; Gan et al., 2023b; Terris & Moreau, 2023; Gossard & Weiss, 2024). It is also worth highlighting the related work that has explored using corrupt measurements for training Ambient DMs (Daras et al., 2023; Aali et al., 2024). These Ambient DMs, trained directly on such undersampled measurements, address the reconstruction problem by sampling from their approximated posterior distribution using a standard diffusion sampler at inference. ShaRP, conversely, adopts an optimization approach, defining a novel objective function from an ensemble of likelihoods of multiple degraded observations.

| Method | Prior Type | Prior Configuration |
|--------|-----------|---------------------|
| PnP | Only Gaussian Denoiser | Single Prior |
| SNORE | Only Gaussian Denoiser | Ensemble of Priors |
| DRP | General Restoration[†] | Single Prior |
| ShaRP | General Restoration[†] | Ensemble of Priors |

[†] Can include denoisers and extends to other restoration tasks.

Table 1: ShaRP differs from existing methods by offering greater flexibility through the use of an ensemble of priors trained on general restoration tasks, extending beyond Gaussian denoising priors.

**Our contribution. (1)** We propose ShaRP, a new framework for solving inverse problems leveraging a set of priors implicit in a pre-trained deep restoration network. As summarized in Table 1, ShaRP generalizes Regularization by Denoising (RED) (Romano et al., 2017) and Stochastic Denoising Regularization (SNORE) (Renaud et al., 2024) by using more flexible restoration operators and generalizes Deep Restoration Priors (DRP) (Hu et al., 2024c) by using

---

**Algorithm 1** Stochastic deep Restoration Priors (ShaRP)

1: **input:** Initial value $\boldsymbol{x}^0 \in \mathbb{R}^n$, $\gamma > 0$, $\sigma > 0$, and $\tau > 0$
2: **for** $k = 1, 2, 3, \dots$ **do**
3: $\quad$ Sample a degradation operator: $\mathbf{H} \sim p(\mathbf{H})$
4: $\quad \boldsymbol{s} \leftarrow \mathbf{H}\boldsymbol{x}^{k-1} + \boldsymbol{n}$ with $\boldsymbol{n} \sim \mathcal{N}(0, \sigma^2 \mathbf{I})$
5: $\quad \boldsymbol{x}^k \leftarrow \boldsymbol{x}^{k-1} - \gamma \widehat{\nabla} f(\boldsymbol{x}^{k-1})$
$\quad\quad$ with $\widehat{\nabla} f(\boldsymbol{x}) \coloneqq \nabla g_{\boldsymbol{A}}(\boldsymbol{x}) + \hat{\nabla} h(\boldsymbol{x})$
$\quad\quad$ where $\widehat{\nabla} h(\boldsymbol{x}) \coloneqq \frac{\tau}{\sigma^2} \mathbf{H}^{\mathsf{T}} \mathbf{H}(\boldsymbol{x} - \mathsf{R}(\boldsymbol{s}, \mathbf{H}))$
6: **end for**

---

multiple restoration priors instead of relying on a single one. **(2)** We introduce a novel regularization concept for inverse problems that encourages solutions that produce degraded versions closely resembling real degraded images. For example, our regularizer favors an MR image solution only if its various degraded versions are consistent with the characteristics of actual degraded MR images. **(3)** We show that ShaRP can be interpreted as a stochastic gradient method that seeks to minimize a composite objective that incorporates our proposed regularizer. We discuss its convergence for both exact and approximate MMSE restoration operators. We also provide theoretical guarantees on learning restoration operators from incomplete measurements. **(4)** We implement ShaRP with both supervised and self-supervised restoration models as priors and test it on two inverse problems: compressed sensing MRI (CS-MRI) and single-image super-resolution (SISR). Our results highlight the capability of restoration models to achieve state-of-the-art performance. Notably, in the MRI context, we show that restoration networks trained directly on subsampled and noisy MRI data can serve as effective priors, a scenario where training traditional Gaussian denoisers is infeasible.

## 3. Stochastic Deep Restoration Priors

ShaRP is presented in Algorithm 1. It considers a prior based on a deep restoration model $\mathsf{R}(\boldsymbol{s}, \mathbf{H})$ pre-trained using the family of degradation operators, such as blur kernels or MRI masks. More specifically, the deep restoration model $\mathsf{R}$

is trained to solve the following set of restoration problems

$$s = \mathbf{H}x + n \quad \text{with} \quad x \sim p_x, \quad \mathbf{H} \sim p(\mathbf{H}), \quad (3)$$

where $n$ is the AWGN vector with variance $\sigma^2$ and $p_x$ denotes the probability distribution of the target images, and $p(\mathbf{H})$ is the probability density of considered degradation operators. Importantly, the restoration problems (3) are used exclusively for training R and do not need to match the target inverse problem (1), which involves the measurement operator $A$.

We prove below that $\widehat{\nabla} f$ corresponds to a stochastic approximation of an objective function of the form $f = g + h$. Here, $g(x) = \frac{1}{2} \|Ax - y\|_2^2$ and the term $h(x)$ is defined as $h(x) = \tau \mathbb{E}_{s \sim G_\sigma(s - \mathbf{H}x), \mathbf{H} \sim p(\mathbf{H})} [-\log p(s|\mathbf{H})]$. We will further introduce the derivation and interpretation of $h(x)$ in Section 4.1. Similar to traditional stochastic gradient methods, ShaRP can be implemented using various selection strategies for the degradation operators.

Each iteration of ShaRP has an intuitive interpretation, where the next solution is obtained by combining the gradient of the data-fidelity term $\nabla g$ and the residual of restored image corresponding to the selected degradation operator. It is worth highlighting that ShaRP is compatible with multiple priors, each restoring an image from Gaussian or non-Gaussian degradations. Also, the computational cost of running ShaRP is comparable to those of single-model approaches. This is due to the stochastic nature of our algorithm that uses a restoration operator in each iteration.

## 4. Theoretical analysis of ShaRP

We now present several theoretical results on ShaRP. Our main contribution is the expression for the closed-form regularizer minimized by ShaRP. We also show how to interpret ShaRP as a stochastic gradient method, enabling its convergence analysis when using inexact MMSE operators. We finally prove theoretical guarantees on learning MMSE restoration operators from unsersampled measurements.

### 4.1. Explicit formula for regularizer

Consider a restoration model that perform MMSE estimation of $x \in \mathbb{R}^n$ for problems (3)

$$
\begin{aligned}
\mathsf{R}^*(s, \mathbf{H}) = \mathbb{E}[x \mid s, \mathbf{H}] &= \int x \, p(x \mid s, \mathbf{H}) \, dx \\
&= \frac{1}{p(s \mid \mathbf{H})} \int x \, G_\sigma(s - \mathbf{H}x) \, p_x(x) \, dx.
\end{aligned}
\quad (4)
$$

where we used the probability density of the observation $s$ conditioned on the operator $\mathbf{H}$

$$p(s|\mathbf{H}) = \int G_\sigma(s - \mathbf{H}x) p_x(x) \, dx. \quad (5)$$

The function $G_\sigma$ in (5) denotes the Gaussian density function with the standard deviation $\sigma > 0$.

We propose the ShaRP regularizer

$$h(x) = \tau \mathbb{E}_{s \sim G_\sigma(s - \mathbf{H}x), \mathbf{H} \sim p(\mathbf{H})} [-\log p(s|\mathbf{H})], \quad (6)$$

where $\tau > 0$ is the regularization parameter and $p(\mathbf{H})$ is the distribution of considered degradations. The regularizer h is minimized when degraded versions of $x$ are highly probable under the distribution $p(s|\mathbf{H})$, with $\mathbf{H}$ sampled from $p(\mathbf{H})$. In other words, a solution $\hat{x}$ is considered effective if its degraded versions $\mathbf{H}\hat{x}$ align with the degraded versions $\mathbf{H}x$ of clean images $x \sim p_x$, for all $\mathbf{H} \sim p(\mathbf{H})$. This can be understood as searching for a fixed point that exhibits equivariance across multiple degradations. The key benefit of the proposed regularizer in (6) lies in its versatility, allowing a wide range of degradation operators to be used in a unified framework. In particular, this formulation remains compatible with Gaussian denoisers, as $\mathbf{H} = \mathbf{I}$ can always be incorporated into $p(\mathbf{H})$.

We are now ready to state our first theoretical result.

**Theorem 1.** *Assume that the prior density $p_x$ is non-degenerate over $\mathbb{R}^n$ and let $\mathsf{R}^*$ be the MMSE restoration operator (4) corresponding to the restoration problems (3). Then, we have that*

$$\nabla h(x) = \frac{\tau}{\sigma^2} \left( \mathbb{E} \left[ \mathbf{H}^\mathsf{T} \mathbf{H}(x - \mathsf{R}^*(s, \mathbf{H})) \right] \right), \quad (7)$$

*where h is the ShaRP regularizer in (6), the expectation is with respect to $s \sim G_\sigma(s - \mathbf{H}x)$ and $\mathbf{H} \sim p_\mathbf{H}$.*

The proof is in the appendix. Note that the expression within the square parenthesis in (7) matches the ShaRP update in Line 4 of Algorithm 1, which directly implies that ShaRP using the exact MMSE restoration operator $\mathsf{R}^*$ is a stochastic gradient method for minimizing $f = g + h$, where $g$ is the data-fidelity term and $h$ is the ShaRP regularizer in (6).

### 4.2. Theoretical convergence of ShaRP

Given the explicit regularized in (6), the iterations of ShaRP in Algorithm 1 can be seen as stochastic gradient method for minimizing $f = g + h$. In practical scenarios, the learned restoration model may be imperfect, meaning it cannot be considered a perfect MMSE estimator. We now present the convergence analysis of ShaRP under a restoration operator R that *approximates* the true MMSE restoration operator $\mathsf{R}^*$. We adopt the analysis of biased stochastic gradient descent (SGD) from the optimization literature (Bertsekas, 2011; Ghadimi & Lan, 2016; Demidovich et al., 2023), by interpreting iterations of ShaRP as a variant of biased SGD. For a given degraded observation $s = \mathbf{H}x + n$ with

$\mathbf{H} \sim p_{\mathbf{H}}$ and $\boldsymbol{n} \sim \mathcal{N}(0, \sigma^2 \mathbf{I})$, we define the stochastic gradient used by ShaRP

$$\widehat{\nabla} f(\boldsymbol{x}) = \nabla g(\boldsymbol{x}) + \widehat{\nabla} h(\boldsymbol{x}),$$

$$\text{where} \quad \widehat{\nabla} h(\boldsymbol{x}) := \frac{\tau}{\sigma^2} \mathbf{H}^{\mathsf{T}} \mathbf{H}\big(\boldsymbol{x} - \mathsf{R}(\boldsymbol{s}, \mathbf{H})\big). \tag{8}$$

Since R is an inexact MMSE restoration operator, we also define the bias vector

$$\boldsymbol{b}(\boldsymbol{x}) = \frac{\tau}{\sigma^2} \mathbb{E}\Big[ \mathbf{H}^{\mathsf{T}} \mathbf{H}\big(\mathsf{R}^*(\boldsymbol{s}, \mathbf{H}) - \mathsf{R}(\boldsymbol{s}, \mathbf{H})\big)\Big] \tag{9}$$

which quantifies the average difference between the exact and inexact MMSE restoration operators with respect to $\mathbf{H} \sim p_{\mathbf{H}}$ and $\boldsymbol{s} \sim G_\sigma(\boldsymbol{s} - \mathbf{H}\boldsymbol{x})$. The analysis requires three assumptions that jointly serve as sufficient conditions for our theorem.

**Assumption 1.** *The function $f$ has a finite minimum $f^* > -\infty$ and the gradient $\nabla f$ is Lipschitz continuous with constant $L > 0$.*

This is a standard assumption used in the analysis of gradient-based algorithms (see (Nesterov, 2004), for example). It is satisfied by a large number of functions, including the traditional least-squares data-fidelity function.

**Assumption 2.** *The stochastic gradient has a bounded variance for all $\boldsymbol{x} \in \mathbb{R}^n$, which means that there exists a constant $\nu > 0$ such that*

$$\mathbb{E}\left[\left\|\widehat{\nabla} f(\boldsymbol{x}) - \mathbb{E}\left[\widehat{\nabla} f(\boldsymbol{x})\right]\right\|_2^2\right] \le \nu^2,$$

*where expectations are with respect to $\mathbf{H} \sim p_{\mathbf{H}}$ and $\boldsymbol{s} \sim G_\sigma(\boldsymbol{s} - \mathbf{H}\boldsymbol{x})$.*

This is another standard assumption extensively used in the analysis of online or stochastic optimization algorithms (Bertsekas, 2011; Ghadimi & Lan, 2016; Demidovich et al., 2023).

**Assumption 3.** *The bias $\boldsymbol{b}(\boldsymbol{x})$, as defined in (9), is bounded, which means that there exists $\varepsilon > 0$ such that for all $\boldsymbol{x} \in \mathbb{R}^n$*

$$\|\boldsymbol{b}(\boldsymbol{x})\|_2 \le \varepsilon.$$

The only assumption on the bias is that it is bounded, which is a relatively mild assumption, as image pixel values are typically constrained (e.g., to [0, 255]).

Note that Assumptions 1-2 are needed only for Proposition 1. They are not needed for our main results—Theorem 1-2.

**Proposition 1.** *Run ShaRP for $t \ge 1$ iterations using the step-size $0 < \gamma \le 1/L$ under Assumptions 1-3. Then, the sequence $\boldsymbol{x}^k$ generated by ShaRP satisfies*

$$\mathbb{E}\left[\frac{1}{t}\sum_{k=1}^{t}\|\nabla f(\boldsymbol{x}^{k-1})\|_2^2\right] \le \frac{2}{\gamma t}(f(\boldsymbol{x}^0) - f^*) + \gamma L \nu^2 + \varepsilon^2.$$

The general analysis of biased stochastic SGD has been extensively discussed in the optimization literature (Bertsekas, 2011; Ghadimi & Lan, 2016; Demidovich et al., 2023). Our contribution here is the relationship between the iterates of ShaRP and those of biased SGD, leading to theoretical guarantees on the stability of ShaRP using restoration networks that do not correspond to ideal MMSE estimators. Proposition 1 states that *in expectation*, ShaRP minimizes the norm of the gradient $\nabla f$ up to an error term that has two components, $\gamma L \nu^2$ and $\epsilon^2$. Since the first component depends on $\gamma$, it can be made as small as desired by controlling the step-size $\gamma$. The second component only depends on the magnitude of the bias $\varepsilon$, which, in turn, directly depends on the accuracy of the restoration operator relative to the true MMSE restoration operator $\mathsf{R}^*$.

### 4.3. Learning restoration priors without groundtruth

In this subsection, we present a theoretical result that establishes the feasibility of learning an MMSE estimator from undersampled measurements. Let the undersampled measurements be defined as $\boldsymbol{s} = \mathbf{H}\boldsymbol{x} + \boldsymbol{n}$, where $\mathbf{H} = \boldsymbol{P}\boldsymbol{M}$. Here, $\boldsymbol{P}$ represents a binary subsampling matrix, $\boldsymbol{M}$ denotes the square transfer operator, and $\boldsymbol{n}$ corresponds to the noise vector.

To show that an MMSE estimator can be learned from undersampled measurement, we need the following assumption.

**Assumption 4.** $\mathbb{E}_{\boldsymbol{P}}[\boldsymbol{P}^{\mathsf{T}}\boldsymbol{P}]$ *has a full rank where the expectation is taken over $p_{\boldsymbol{P}}$. The measurement operator $\boldsymbol{M}$ is orthogonal matrix.*

This assumption implies that the *union* of all sampling matrices $\boldsymbol{P}$ spans the complete measurement domain, even though each individual $\boldsymbol{P}$ may remain undersampled.

**Theorem 2.** *Under Assumption 4, the MMSE estimator R learned using the weighted self-supervised loss ($\ell_{\mathsf{self}}$) is equivalent to its supervised counterpart ($\ell_{\mathsf{sup}}$). Specifically, we have:*

$$\mathsf{R}_{\ell_{\mathsf{self}}}(\boldsymbol{\theta}) = \mathsf{R}_{\ell_{\mathsf{sup}}}(\boldsymbol{\theta}) . \tag{10}$$

*where*

$$\ell_{\mathsf{sup}} = \mathbb{E}\left[\frac{1}{2}\|\overline{\boldsymbol{x}} - \boldsymbol{x}\|_2^2\right] \tag{11}$$

*and*

$$\ell_{\mathsf{self}} = \mathbb{E}\left[\frac{1}{2}\|\boldsymbol{P}'\boldsymbol{M}\overline{\boldsymbol{x}} - \boldsymbol{s}'\|_{\boldsymbol{W}}^2\right], \tag{12}$$

*where the vector $\overline{\boldsymbol{x}} = \mathsf{R}(\boldsymbol{s})$ is MMSE estimation of R for $\boldsymbol{s}$, $\boldsymbol{s}' = \boldsymbol{P}'\boldsymbol{M}\boldsymbol{x}$ is another independently subsampled measurement and $\boldsymbol{W}$ is a weighting matrix to compensate sampling imbalance in measurement domain.*

## 5. Numerical Results

We numerically validate ShaRP on two inverse problems of the form $\boldsymbol{y} = \boldsymbol{A}\boldsymbol{x}+\boldsymbol{e}$: (*Compressive Sensing MRI (CS-MRI)* and (b) *Single Image Super Resolution (SISR)*. In both cases, $\boldsymbol{e}$ represents additive white Gaussian noise (AWGN). For the data-fidelity term in eq. (2), we use the $\ell_2$-norm loss for both problems. Quantitative performance is evaluated by Peak Signal-to-Noise Ratio (PSNR) and Structural Similarity Index (SSIM). Additionally, for the SISR task, we include the Learned Perceptual Image Patch Similarity (LPIPS) metric to evaluate perceptual quality. Additional numerical results are provided in the supplementary material.

| Noise level | $\sigma = 0.005$ | | $\sigma = 0.010$ | | $\sigma = 0.015$ | |
|---|---|---|---|---|---|---|
| Metrics | PSNR | SSIM | PSNR | SSIM | PSNR | SSIM |
| Zero-filled | 26.93 | 0.848 | 26.92 | 0.847 | 26.90 | 0.848 |
| TV | 31.17 | 0.923 | 31.08 | 0.921 | 30.91 | 0.915 |
| PnP-FISTA | 35.88 | 0.938 | 31.14 | 0.894 | 30.32 | 0.846 |
| PnP-ADMM | 35.76 | 0.941 | 32.36 | 0.878 | 30.66 | 0.838 |
| DRP | 35.52 | 0.936 | 32.32 | 0.914 | 30.57 | 0.901 |
| DPS | 32.62 | 0.888 | 31.39 | 0.870 | 30.29 | 0.856 |
| DDS | 35.21 | 0.937 | 35.03 | 0.935 | 34.51 | 0.925 |
| ShaRP | **37.59** | **0.963** | **35.81** | **0.951** | **34.92** | **0.942** |
| E2E-VarNet[†] | 38.10 | 0.971 | 36.80 | 0.967 | 35.79 | 0.954 |

[†]E2E-VarNet needs to be retrained for each task.

Table 2: Quantitative comparison of ShaRP with several baselines for CS-MRI using $4\times$ uniform masks. The **best** and second best results across general image restoration methods are highlighted. Notably, ShaRP outperforms SOTA baseline methods.

| Noise level | $\sigma = 0.005$ | | $\sigma = 0.010$ | | $\sigma = 0.015$ | |
|---|---|---|---|---|---|---|
| Metrics | PSNR | SSIM | PSNR | SSIM | PSNR | SSIM |
| ADMM-TV | 28.14 | 0.866 | 28.06 | 0.863 | 27.96 | 0.859 |
| GRAPPA | 28.09 | 0.792 | 25.39 | 0.699 | 23.94 | 0.649 |
| SPICER | 31.87 | 0.901 | 31.67 | 0.889 | 31.50 | 0.887 |
| ShaRP[self] | **33.87** | **0.909** | **33.64** | **0.900** | **33.21** | **0.892** |

Table 3: Quantitative comparison of ShaRP with a self-supervised pre-trained restoration operator, compared to several baselines for CS-MRI using $4\times$ uniform masks. The **best** and second best results are highlighted. Note the excellent performance of ShaRP even using priors trained without fully-sampled ground-truth data.

### 5.1. CS-MRI setting

The complex-valued measurement model for CS-PMRI is expressed as: $\boldsymbol{y} = \boldsymbol{P}\boldsymbol{F}\boldsymbol{S}\boldsymbol{x} + \boldsymbol{e}$. Here, $\boldsymbol{x}$ represents the underlying complex-valued image, $\boldsymbol{S}$ are the complex multi-coil sensitivity maps, $\boldsymbol{F}$ is the Fourier transform operator,

and $\boldsymbol{P}$ is the k-space subsampling mask. The term $\boldsymbol{y}$ denotes the acquired complex k-space data, and $\boldsymbol{e}$ is the additive complex noise vector. We utilized the open-access fastMRI dataset; further experimental details can be found in Section B.1 of the supplementary material.

**Ensemble of restoration priors for CS-MRI.** Recent methods like InDI (Delbracio & Milanfar, 2023) and $I^2SB$ (Liu et al., 2023) use controllable processes to train ensembles of restoration priors, each an MMSE operator for a specific setting. We build on this by training an $8\times$ uniform subsampling CS-MRI model with 8 masks as our prior. Following InDI, we decompose the MRI degradation operator $\boldsymbol{M} = \boldsymbol{P}\boldsymbol{F}\boldsymbol{S}$ into convex combinations of $\boldsymbol{M}$ and the identity $\mathbf{I}$: $\mathbf{H}_\alpha = (1 - \alpha)\mathbf{I} + \alpha\boldsymbol{M}$, with $\alpha$ controlling degradation. Varying $\alpha$ creates an ensemble of tasks. Training R on all these tasks allows it to act as an ensemble of MMSE operators: $\mathsf{R}(\boldsymbol{s}, \mathbf{H}_\alpha) = \mathbb{E}\left[\boldsymbol{x}|\boldsymbol{s}, \mathbf{H}_\alpha\right]$. We used the MSE loss for training.

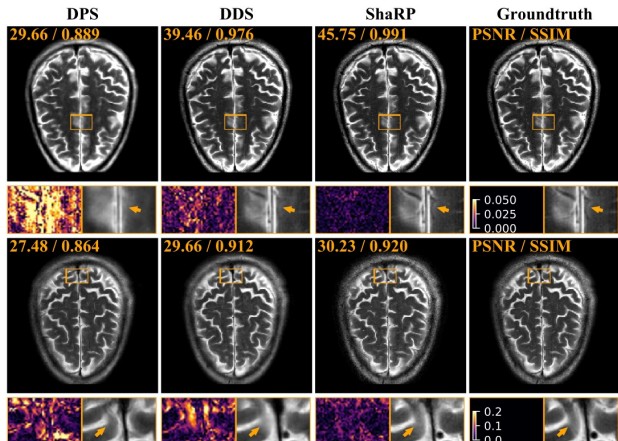

Figure 2: Visual comparison of ShaRP with baseline methods on CS-MRI. The top row shows results for a $4\times$ random mask with noise $\sigma = 0.005$, and the bottom row for a $6\times$ random mask with noise $\sigma = 0.015$. PSNR and SSIM values are in the top-left corner of each image. Error maps and zoomed-in areas highlight differences. Notably, ShaRP with stochastic priors outperforms state-of-the-art methods using denoiser and diffusion model priors.

**Training restoration priors without groundtruth.** When fully-sampled images are unavailable for training, MRI restoration priors can be trained self-supervisedly (Yaman et al., 2020; Millard & Chiew, 2023; Gan et al., 2023b; Hu et al., 2024d), as demonstrated in Theorem 2. Notably, the MRI imaging system aligns well with the assumptions required by Theorem 2, as $\boldsymbol{F}\boldsymbol{S}$ represents an orthogonal matrix. This approach uses a separate subsampled measurement as the training label, instead of the ground-truth image. Importantly, and consistent with Theorem 2, our self-supervised training *theoretically* achieves MMSE restora-

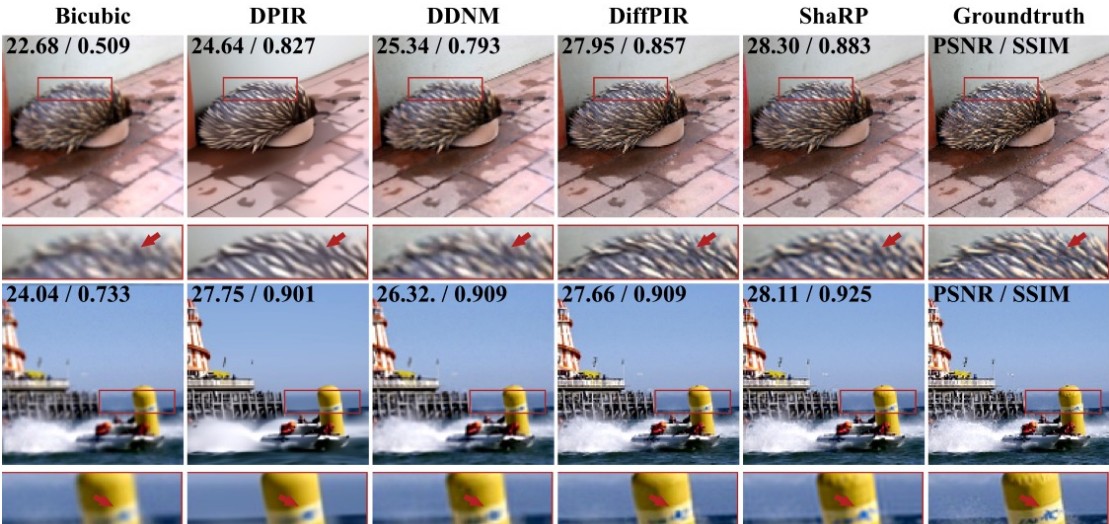

Figure 3: Visual comparison of ShaRP with several well-known methods on $2\times$ SISR. The top row shows results for SISR with gaussian blur kernel with $\sigma = 1.25$, while the bottom row shows results for SISR with gaussian blur kernel with $\sigma = 1.5$. The quantities in the top-left corner of each image provide PSNR and SSIM values for each method. The squares at the bottom of each image visualize the zoomed area in the image.

tion priors equivalent to those trained fully-supervisedly. For validation, we train an $8\times$ uniform subsampling CS-MRI model to handle eight distinct restoration operators corresponding to different sampling masks. Additional training details are in Section B.1 of the supplement.

With the pre-trained $8\times$ models as ensembles of restoration priors, we evaluate ShaRP's performance across a variety of configurations, including two sub-sampling rates ($4\times$ and $6\times$), two mask types (uniform and random), and three noise levels ($\sigma = 0.005$, $0.01$, and $0.015$). Due to space constraints, only a subset of these settings is included in the main paper. Additional results can be found in Section B.1 of the supplementary material.

**Baselines.** ShaRP was compared against several baseline methods, including denoiser-based approaches (PnP-FISTA (Kamilov et al., 2017), PnP-ADMM (Chan et al., 2017)) and diffusion model-based methods (DPS (Chung et al., 2023), DDS (Chung et al., 2024)). To highlight the advantages of using a stochastic set of restoration operators, we also compared ShaRP with the DRP method (Hu et al., 2024c), which applies only a single restoration operator. Additional details related to the baseline methods can be found in Section B.1 of the appendix.

**Results with supervised MMSE restoration operator.** Table 2 provides a quantitative comparison of reconstruction performance across different acceleration factors and noise levels using a uniform sub-sampling mask. In all configurations, ShaRP consistently outperforms the baseline methods. The use of a set of restoration operators clearly enhances

ShaRP's performance, highlighting the effectiveness of employing multiple operators to maximize the regularization information provided by the restoration model. Figure 2 presents visual reconstructions for two test scenarios, where ShaRP accurately recovers fine brain details, particularly in the zoomed-in regions, while baseline methods tend to oversmooth or introduce artifacts. These results highlight ShaRP's superior ability to manage structured artifacts and preserve fine details, outperforming both denoiser-based and diffusion model-based methods.

**Results with MMSE restoration operator learned from incomplete measurements.** We further evaluate ShaRP's performance using an restoration model, learned in a self-supervised manner, as introduced in (Gan et al., 2023b). In this setting, we compare ShaRP against two classical methods for CS-MRI reconstruction without groundtruth: TV (Block et al., 2007) and GRAPPA (Griswold et al., 2002) and a recent state-of-the-art self-supervised deep unrolling method: SPICER (Hu et al., 2024d). As shown in Table 3, ShaRP demonstrates its effectiveness in leveraging a self-supervised restoration prior for various reconstruction tasks, even when only incomplete measurements ($8\times$ subsampled) are available. Note that, ShaRP using self-supervised restoration prior even outperforms DPIR and DPS that use Gaussian denoisers trained using fully-sampled ground truth images (see Table 6 in the appendix). It is important to highlight that training Gaussian denoisers is infeasible when only undersampled measurements are available.

We further evaluate the convergence performance of ShaRP using both supervised and self-supervised restoration priors.

Due to space limitations, the detailed results and discussion are provided in Section C.3.

## 5.2. Single Image Super Resolution (SISR)

The measurement operator in SISR can be written as $A = SK$, where $K$ represents convolution with the blur kernel, and $S$ performs standard $d$-fold down-sampling. In our experiments, we use two Gaussian blur kernels $k$ , each with distinct standard deviations (1.25 and 1.5), and with down-sampling factor of 2. Both noisy and noise-free cases are considered to evaluate the noisy robustness of ShaRP. We randomly selected 100 images from the ImageNet test set, as provided in DiffPIR[1]. Due to space constraints, only a subset of comparisons is included in the main paper. Additional results can be found in Section D of the supplement.

**Ensemble of Restoration Priors for Image Deblurring.** Similar to our CS-MRI prior training, we decompose the deblurring task (Gaussian blur operator $K$) into a convex combination of $K$ and the identity mapping I: $\mathbf{H}_\alpha = (1 - \alpha)\mathsf{I} + \alpha K$, with $\alpha$ controlling degradation. Varying $\alpha$ yields multiple degradation operators, enabling our restoration network R to handle each. This allows R to function as an ensemble of MMSE restoration operators: $\mathsf{R}(s, \mathbf{H}_\alpha) = \mathbb{E}[x \mid s, \mathbf{H}_\alpha]$, where $s$ and $x$ are the degraded and original images, respectively. The original $K$ uses a $31 \times 31$ Gaussian kernel with a standard deviation of 3. Further training details are in Section B.2 of the supplement.

**Baselines.** We compared ShaRP against baselines including DPIR (Zhang et al., 2022), a state-of-the-art PnP method using pre-trained denoisers, and diffusion-based methods DPS (Chung et al., 2023), DDNM (Wang et al., 2023), DDRM(Kawar et al., 2022), and DiffPIR(Zhu et al., 2023), which use different sampling strategies for SISR.

**Results on SISR with deblurring prior.** Figure 3 shows the visual reconstruction results for two settings with different blur kernels. As demonstrated, ShaRP successfully recovers most features and maintains high data consistency with the available measurements. Table 4 provides quantitative comparisons against baselines across blur kernels and noise levels. ShaRP achieves the highest PSNR and SSIM, but ranks second in perceptual performance (LPIPS), consistent with SOTA Diffusion Model-based methods on SISR. However, the deblurring prior in ShaRP recovers fine details, ensuring competitive perceptual quality. One interpretation for the improved performance is that when considering an ensemble of blur-then-deblur operations, ShaRP can be seen as masking and subsequently restoring portions of null-space—akin to the mechanism of masked (denoising) autoencoders. In this interpretation, MMSE restoration networks effectively reconstruct the missing null-space signals,

which improves the perceptual quality. Additional results can be found in Section D of supplementary material.

## 6. Discussion

**ShaRP as a Unifed Framework.** Unlike methods that rely on a fixed prior for all tasks, ShaRP operates without being constrained to any specific prior. Instead, it adapts flexibly to the unique characteristics of diverse inverse problems, enabling the integration of a broad range of restoration priors, including—but not limited to—denoisers. This adaptability not only enhances the versatility and applicability of the framework but also has the potential to achieve superior performance, as demonstrated in Section 5. By building on a unified theoretical foundation, ShaRP represents a significant advancement in addressing a wide variety of inverse problems effectively.

**Performance improved with Ensemble Priors.** Conventional ensemble priors for inverse problems typically focus on Gaussian denoisers across varying noise levels (Zhang et al., 2019; Hurault et al., 2022a; Renaud et al., 2024). ShaRP advances this concept by integrating structured noise from diverse degradation operators, forming a richer and more robust ensemble. Empowered by its flexible and novel regularizer, ShaRP seamlessly incorporates this diverse ensemble into a unified, closed-form prior that accommodates both Gaussian and more complex denoisers. This innovation highlights ShaRP's ability to leverage a broader range of priors, enabling adaptable and robust performance across a wide variety of noise environments. Further evidence of the benefits of such an expansive ensemble is provided by an ablation study in Section E.2.

ShaRP's ability to integrate diverse restoration priors within a unified framework enables task-specific adaptability, driving significant performance improvements and advancing inverse problem-solving.

**Sampling from Restoration Priors.** Our current work introduces a novel objective function balancing data fidelity with implicit regularization from an ensemble of priors. Although this optimization-based approach achieves high reconstruction accuracy, its perceptual performance is inherently limited by the well-known perceptual-distortion trade-off (Blau & Michaeli, 2018). To enhance the perceptual performance, sampling-based extension is a promising future direction. Specifically, we aim to adapt existing diffusion-based image restoration methods (Daras et al., 2023; Bansal et al., 2023; Delbracio & Milanfar, 2023; Liu et al., 2023), initially developed for specific tasks, to serve as priors for sampling in general image restoration.

---

[1]https://github.com/yuanzhi-zhu/DiffPIR/tree/main/testsets

| Noise level | Noiseless | | | $\sigma = 0.01$ | | | Noiseless | | | $\sigma = 0.01$ | | |
|---|---|---|---|---|---|---|---|---|---|---|---|---|
| Metrics | PSNR | SSIM | LPIPS | PSNR | SSIM | LPIPS | PSNR | SSIM | LPIPS | PSNR | SSIM | LPIPS |
| DPIR | 28.10 | 0.809 | 0.305 | 28.05 | 0.807 | 0.308 | 27.90 | 0.803 | 0.314 | 27.87 | 0.800 | 0.314 |
| DDNM | 27.53 | 0.786 | 0.240 | 27.49 | 0.784 | 0.246 | 27.02 | 0.764 | 0.264 | 27.01 | 0.763 | 0.267 |
| DPS | 24.68 | 0.661 | 0.395 | 24.60 | 0.657 | 0.399 | 24.50 | 0.657 | 0.403 | 24.44 | 0.655 | 0.406 |
| DiffPIR | 28.92 | 0.852 | **0.152** | 28.63 | 0.839 | **0.169** | 28.59 | 0.834 | **0.172** | 28.02 | 0.819 | **0.185** |
| DDRM | 28.20 | 0.845 | 0.161 | 28.11 | 0.832 | 0.188 | 27.93 | 0.826 | 0.188 | 27.67 | 0.817 | 0.193 |
| DRP | 29.28 | 0.868 | 0.207 | 28.87 | 0.848 | 0.248 | 28.24 | 0.836 | 0.235 | 28.01 | 0.820 | 0.278 |
| ShaRP | **30.09** | **0.891** | 0.179 | **29.03** | **0.852** | 0.223 | **29.28** | **0.872** | 0.209 | **28.06** | **0.821** | 0.268 |

Table 4: Quantitative comparison of ShaRP with several baselines for SISR based on two different blur kernels on ImageNet dataset. The **best** and second best results are highlighted. Notably, ShaRP outperforms SOTA methods based on denoisers and diffusion models.

## 7. Conclusion

This work presents ShaRP, a novel framework for imaging inverse problems that leverages pre-trained restoration networks as priors. Unlike methods limited to Gaussian denoisers or a single prior, ShaRP integrates multiple priors tailored to diverse degradations within a unified theoretical framework. The key findings are: (1) using priors beyond traditional Gaussian denoisers broadens the framework's applicability and performance, and (2) stochastically integrating multiple degradation-specific priors achieves better performance than relying on a single prior. Numerical experiments confirm ShaRP's superiority over conventional methods. By unifying diverse priors into a flexible framework, ShaRP encourages exploring more complex restoration priors for inverse problem-solving.

## Ethics Statement

To the best of our knowledge this work does not give rise to any significant ethical concerns.

## Impact Statement

ShaRP makes advanced computational imaging more accessible by reducing reliance on task-specific models and extensive supervision. Its versatility enables broad societal impact across fields that depend on high-quality image data, such as healthcare, scientific research, and consumer technology.

## Acknowledgement

Research presented in this article was supported in part by the NSF CAREER awards under grant CCF-2043134.

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

**Here's what we cover in the appendix:**

- **Theoretical Analysis of ShaRP**

  - Proof of Theorem 1 (detailed in Subsection A.1): Derivation of the gradient of the ShaRP regularizer.
  - Proof of Proposition 1 (detailed in Subsection A.2): Convergence guarantees for ShaRP with inexact Minimum Mean Square Error (MMSE) restoration operators using biased Stochastic Gradient Descent (SGD) analysis.
  - Proof of Theorem 2 (detailed in Subsection A.3): Equivalence of the MMSE estimator learned via weighted self-supervised loss to its supervised counterpart.

- **Experiment Details** (Section B)

  - Implementation details for Compressed Sensing MRI (CS-MRI) tasks (Subsection B.1):
  - Implementation details for Single Image Super-Resolution (SISR) tasks (Subsection B.2):

- **Additional results for CS-MRI** (Section C)

  - Quantitative performance of ShaRP for uniform (Table 5) and random (Table 6) subsampling settings.
  - Visual comparisons for CS-MRI tasks (Figure 5, Figure 6).
  - Performance of ShaRP with self-supervised restoration priors (Subsection C.2, Table 7).
  - Convergence performance analysis of ShaRP with both supervised and self-supervised priors (Subsection C.3, Figure 7), aligning with the convergence analysis for inexact MMSE estimators.
  - Performance comparison with additional baseline methods on matched and mismatched settings (Table 8, Figure 8).

- **Additional visual results for SISR** (Section D)

  - Further visual comparisons against various baseline methods for SISR (Figure 9, Figure 10).
  - Additional visual comparisons against the DRP method (Figure 11).
  - Quantitative (Table 10) and visual (Figure 12) comparisons against additional diffusion-based baselines.

- **Additional Experiments**

  - Ablation study on using a pre-trained super-resolution prior for the CS-MRI task (Subsection E.1, Algorithm 5, Table 12).
  - Ablation study on the impact of the number of restoration priors, $b$, in the ensemble on CS-MRI performance (Subsection E.2, Figure 13, Figure 14).
  - Ablation study on the influence of the hyperparameter $\alpha$ (controlling prior selection, as introduced in Subsection B.1) on CS-MRI performance (Subsection E.3, Figure 15).

## A. Theoretical Analysis of ShaRP

### A.1. Proof of Theorem 1

**Theorem.** *Assume that the prior density $p_{\boldsymbol{x}}$ is non-degenerate over $\mathbb{R}^n$ and let $\mathsf{R}^*$ be the MMSE restoration operator* (4) *corresponding to the restoration problems* (3). *Then, we have that*

$$\nabla h(\boldsymbol{x}) = \frac{\tau}{\sigma^2} \left( \mathbb{E}_{\boldsymbol{s}\sim G_\sigma(\boldsymbol{s}-\mathbf{H}\boldsymbol{x}), \mathbf{H}\sim p_{\mathbf{H}}} \left[ \mathbf{H}^\mathsf{T} \mathbf{H}(\boldsymbol{x} - \mathsf{R}^*(\boldsymbol{s}, \mathbf{H})) \right] \right),$$

*where $h$ is the ShaRP regularizer in* (6).

*Proof.* The ShaRP regularizer $h(\boldsymbol{x})$ is defined as

$$
\begin{aligned}
h(\boldsymbol{x}) &= \tau \mathbb{E}_{\boldsymbol{s}\sim G_\sigma(\boldsymbol{s}-\mathbf{H}\boldsymbol{x}), \mathbf{H}\sim p_{\mathbf{H}}} \left[ -\log p(\boldsymbol{s}|\mathbf{H}) \right] \\
&= -\tau \int p(\mathbf{H}) \left[ \int G_\sigma(\boldsymbol{s} - \mathbf{H}\boldsymbol{x})\log p(\boldsymbol{s}|\mathbf{H})\, \mathrm{d}\boldsymbol{s} \right] \, \mathrm{d}\mathbf{H},
\end{aligned}
\tag{13}
$$

where $G_\sigma$ is the Gaussian probability density with variance $\sigma^2$ and $p(s|\mathbf{H})$ is the likelihood function for the degraded observation given the operator $\mathbf{H}$. The expectation over $p(\mathbf{H})$ accounts for the randomness of the restoration operator $\mathbf{H}$.

We start by relating the MMSE restoration operator to the score of the degraded observation

$$\nabla p(s|\mathbf{H}) = \frac{1}{\sigma^2} \int (\mathbf{H}x - s) \, G_\sigma(s - \mathbf{H}x) p_x(x) \, \mathrm{d}x,$$

where $p_x$ is the prior. By using the definition of the MMSE estimator, we obtain the relationship

$$\nabla \mathrm{log} p(s|\mathbf{H}) = \frac{1}{\sigma^2} \left( \mathbf{H} \mathsf{R}^*(s, \mathbf{H}) - s \right). \tag{14}$$

Consider the function inside the parenthesis in the expression for the ShaRP regularizer (13)

$$\rho(z) := (G_\sigma * \mathrm{log} p_{s|\mathbf{H}})(z) = \int G_\sigma(z - s) \, \mathrm{log} p(s|\mathbf{H}) \, \mathrm{d}s,$$

where $z$ has the same dimensions as $s$ and $*$ denotes convolution. The gradient of $\rho$ is given by

$$\begin{aligned}
\nabla \rho(z) &= (\nabla G_\sigma * \mathrm{log} p_{s|\mathbf{H}})(z) = (G_\sigma * \nabla \mathrm{log} p_{s|\mathbf{H}})(z) \\
&= \frac{1}{\sigma^2} \int G_\sigma(z - s) \left[ \mathbf{H} \mathsf{R}^*(s, \mathbf{H}) - s \right] \mathrm{d}s \\
&= \frac{1}{\sigma^2} \left( \mathbf{H} \int \mathsf{R}^*(s, \mathbf{H}) G_\sigma(z - s) \, \mathrm{d}s - z \right)
\end{aligned}$$

where we used (14). By using $z = \mathbf{H}x$, we write the gradient with respect to $x$

$$\nabla_x \rho(\mathbf{H}x) = \frac{1}{\sigma^2} \mathbf{H}^\mathsf{T} \mathbf{H} \left( \int \mathsf{R}^*(s, \mathbf{H}) G_\sigma(s - \mathbf{H}x) \, \mathrm{d}s - x \right)$$

By using this expression in (13), we obtain the desired result

$$\begin{aligned}
\nabla h(x) &= -\frac{\tau}{\sigma^2} \left[ \int p(\mathbf{H}) \int G_\sigma(s - \mathbf{H}x) \left( \mathbf{H}^\mathsf{T} \mathbf{H}(\mathsf{R}^*(s, \mathbf{H}) - x) \right) \, \mathrm{d}s \, \mathrm{d}\mathbf{H} \right] \\
&= \frac{\tau}{\sigma^2} \mathbb{E}_{s \sim G_\sigma(s - \mathbf{H}x), \mathbf{H} \sim p_\mathbf{H}} \left[ \mathbf{H}^\mathsf{T} \mathbf{H}(x - \mathsf{R}^*(s, \mathbf{H})) \right].
\end{aligned}$$

$\square$

### A.2. Proof of Proposition 1

We adopt the analysis of biased stochastic gradient descent (SGD) from the optimization literature (Bertsekas, 2011; Ghadimi & Lan, 2016; Demidovich et al., 2023) to provide theoretical convergence guarantees to ShaRP under inexact MMSE restoration operators. Our contribution here is the interpretation of the iterations of ShaRP as a variant of biased SGD, rather than a general analysis of SGD, which is well-known in the optimization literature.

**Proposition.** *Run ShaRP for $t \geq 1$ iterations using the step-size $0 < \gamma \leq 1/L$ under Assumptions 1-3. Then, the sequence $x^k$ generated by ShaRP satisfies*

$$\mathbb{E} \left[ \frac{1}{t} \sum_{k=1}^{t} \|\nabla f(x^{k-1})\|_2^2 \right] \leq \frac{2}{t} (f(x^0) - f^*) + \gamma L \nu^2 + \varepsilon^2.$$

*Proof.* First note that from the definition of the bias in eq. (9), we have that

$$\mathbb{E} \left[ \widehat{\nabla} f(x^{k-1}) \,|\, x^{k-1} \right] = \nabla f(x^{k-1}) + b(x^{k-1}), \tag{15}$$

where the expectation is with respect to $s \sim G_\sigma(s - \mathbf{H}x^{k-1})$ and $\mathbf{H} \sim p_\mathbf{H}$. In order to simplify the notation, we will drop these subscripts from the expectations in the analysis below.

Consider the iteration $k \geq 1$ of ShaRP with inexact MMSE operator

$$f(\boldsymbol{x}^k) \leq f(\boldsymbol{x}^{k-1}) + \nabla f(\boldsymbol{x}^{k-1})^\mathsf{T}(\boldsymbol{x}^k - \boldsymbol{x}^{k-1}) + \frac{L}{2}\|\boldsymbol{x}^k - \boldsymbol{x}^{k-1}\|_2^2$$

$$= f(\boldsymbol{x}^{k-1}) - \gamma\nabla f(\boldsymbol{x}^{k-1})^\mathsf{T}\widehat{\nabla}f(\boldsymbol{x}^{k-1}) + \frac{L\gamma^2}{2}\|\widehat{\nabla}f(\boldsymbol{x}^{k-1})\|^2,$$

where in the first line we used the Lipschitz continuity of $\nabla f$. By taking the expectation with respect to $\boldsymbol{s} \sim G_\sigma(\boldsymbol{s} - \mathbf{H}\boldsymbol{x}^{k-1})$ and $\mathbf{H} \sim p_\mathbf{H}$ on both sides of this expression, we get

$$\mathbb{E}[f(\boldsymbol{x}^k)|\boldsymbol{x}^{k-1}] \leq f(\boldsymbol{x}^{k-1}) - \gamma\nabla f(\boldsymbol{x}^{k-1})^\mathsf{T}(\nabla f(\boldsymbol{x}^{k-1}) + \boldsymbol{b}(\boldsymbol{x}^{k-1})) + \frac{L\gamma^2}{2}\mathbb{E}\left[\|\widehat{\nabla}f(\boldsymbol{x}^{k-1})\|_2^2|\boldsymbol{x}^{k-1}\right]$$

$$\leq f(\boldsymbol{x}^{k-1}) - \frac{\gamma}{2}\|\nabla f(\boldsymbol{x}^{k-1})\|_2^2 + \frac{\gamma}{2}\|\boldsymbol{b}(\boldsymbol{x}^{k-1})\|_2^2$$

$$+ \frac{L\gamma^2}{2}\left(\mathbb{E}\left[\|\widehat{\nabla}f(\boldsymbol{x}^{k-1})\|_2^2|\boldsymbol{x}^{k-1}\right] - \left(\mathbb{E}[\widehat{\nabla}f(\boldsymbol{x}^{k-1})|\boldsymbol{x}^{k-1}]\right)^2\right)$$

$$\leq f(\boldsymbol{x}^{k-1}) - \frac{\gamma}{2}\|\nabla f(\boldsymbol{x}^{k-1})\|_2^2 + \frac{\gamma\varepsilon^2}{2} + \frac{L\gamma^2\nu^2}{2}.$$

In the second row, we completed the square, applied eq. (15), and used the assumption that $\gamma \leq 1/L$. In the third row, we used the variance and bias bounds in Assumptions 2 and 3. By rearranging the expression, we get the following bound

$$\|\nabla f(\boldsymbol{x}^{k-1})\|_2^2 \leq \frac{2}{\gamma}\left(f(\boldsymbol{x}^{k-1}) - \mathbb{E}[f(\boldsymbol{x}^k)|\boldsymbol{x}^{k-1}]\right) + L\gamma\nu^2 + \varepsilon^2$$

By taking the total expectation, averaging over $t$ iterations, and using the lower bound $f^*$, we get the desired result

$$\mathbb{E}\left[\frac{1}{t}\sum_{k=1}^{t}\|\nabla f(\boldsymbol{x}^{k-1})\|_2^2\right] \leq \frac{2}{\gamma t}(f(\boldsymbol{x}^0) - f^*) + L\gamma\nu^2 + \varepsilon^2.$$

$\square$

## A.3. Proof of Theorem 2:

Let the undersampled measurements be defined as $\boldsymbol{s} = \mathbf{H}\boldsymbol{x} + \boldsymbol{n}$, where $\mathbf{H} = \boldsymbol{P}\boldsymbol{M}$. Here, $\boldsymbol{P}$ represents a binary subsampling matrix, $\boldsymbol{M}$ denotes the square transfer operator, and $\boldsymbol{n}$ corresponds to the noise vector.

To show that an MMSE estimator can be learned from undersampled measurement, we need the following assumption.

**Assumption 5.** $\mathbb{E}_{\boldsymbol{P}}[\boldsymbol{P}^\mathsf{T}\boldsymbol{P}]$ has a full rank and $\boldsymbol{M}$ is an orthogonal matrix, where the expectation is taken over $p_{\boldsymbol{P}}$.

This assumption implies that the *union* of all sampling matrices $\boldsymbol{P}$ spans the complete measurement domain, even though each individual $\boldsymbol{P}$ may remain undersampled. This property can be achieved by incorporating an additional weight $\boldsymbol{W}$ into the loss function, where: $\boldsymbol{W} = \boldsymbol{P}'\overline{\boldsymbol{W}}(\boldsymbol{P}'\overline{\boldsymbol{W}})^\mathsf{T} \in \mathbb{R}^{m \times m}$ denotes a subsampled variant of $\overline{\boldsymbol{W}}$ given $\boldsymbol{P}'$.

**Proposition 2.** *When Assumption 2 is satisfied,*

$$\mathbb{E}\left[\boldsymbol{M}^\mathsf{T}\boldsymbol{P}'^\mathsf{T}\boldsymbol{W}\boldsymbol{P}'\boldsymbol{M}\right] = \boldsymbol{I},$$

*where the expectation is with respect to $p_{\boldsymbol{M}}$. This proof is the same as provided in previous work (Gan et al., 2023b).*

**Theorem 3.** *Under Assumption 5, the MMSE estimator* R *learned using the weighted self-supervised loss ($\ell_{\mathsf{self}}$) is equivalent to its supervised counterpart ($\ell_{\mathsf{sup}}$). Specifically, we have:*

$$\mathsf{R}_{\ell_{\mathsf{self}}}(\boldsymbol{\theta}) = \mathsf{R}_{\ell_{\mathsf{sup}}}(\boldsymbol{\theta}). \tag{16}$$

*where*

$$\ell_{\mathsf{sup}} = \mathbb{E}\left[\frac{1}{2}\|\overline{\boldsymbol{x}} - \boldsymbol{x}\|_2^2\right] \tag{17}$$

*and*

$$\ell_{\text{self}} = \mathbb{E}\left[\frac{1}{2}\|\boldsymbol{P}'\boldsymbol{M}\overline{\boldsymbol{x}} - \boldsymbol{s}'\|_{\boldsymbol{W}}^2\right]. \tag{18}$$

*The vector $\overline{\boldsymbol{x}} = \mathsf{R}(\boldsymbol{s})$ is MMSE estimation of $\mathsf{R}$ for $\boldsymbol{s}$.*

*Proof.* Note that the self-supervised loss involves the term $\boldsymbol{P}'\boldsymbol{M}\bar{\boldsymbol{x}} - \boldsymbol{s}'$, where $\boldsymbol{s}' = \boldsymbol{P}'\boldsymbol{M}\boldsymbol{x} + \boldsymbol{n}'$

$$\boldsymbol{P}'\boldsymbol{M}\bar{\boldsymbol{x}} - \boldsymbol{s}' = \boldsymbol{P}'\boldsymbol{M}(\bar{\boldsymbol{x}} - \boldsymbol{x}) - \boldsymbol{n}'. \tag{19}$$

Thus, the self-supervised loss becomes:

$$\ell_{\text{self}} = \mathbb{E}\left[\frac{1}{2}\|\boldsymbol{P}'\boldsymbol{M}(\bar{\boldsymbol{x}} - \boldsymbol{x}) - \boldsymbol{n}'\|_{\boldsymbol{W}}^2\right]. \tag{20}$$

Expanding the squared term:

$$\begin{aligned}
\|\boldsymbol{P}'\boldsymbol{M}(\bar{\boldsymbol{x}} - \boldsymbol{x}) - \boldsymbol{n}'\|_{\boldsymbol{W}}^2 &= \|\boldsymbol{P}'\boldsymbol{M}(\bar{\boldsymbol{x}} - \boldsymbol{x})\|_{\boldsymbol{W}}^2 - 2(\boldsymbol{P}'\boldsymbol{M}(\bar{\boldsymbol{x}} - \boldsymbol{x}))^{\mathsf{T}}\boldsymbol{W}\boldsymbol{n}' + \|\boldsymbol{n}'\|_{\boldsymbol{W}}^2 \\
&= (\bar{\boldsymbol{x}} - \boldsymbol{x})^{\mathsf{T}}\boldsymbol{M}^{\mathsf{T}}\boldsymbol{P}'^{\mathsf{T}}\boldsymbol{W}\boldsymbol{P}'\boldsymbol{M}'(\bar{\boldsymbol{x}} - \boldsymbol{x}) - 2(\boldsymbol{P}'\boldsymbol{M}(\bar{\boldsymbol{x}} - \boldsymbol{x}))^{\mathsf{T}}\boldsymbol{W}\boldsymbol{n}' + \|\boldsymbol{n}'\|_{\boldsymbol{W}}^2.
\end{aligned}$$

So that

$$\begin{aligned}
&\mathbb{E}\left[\|\boldsymbol{P}'\boldsymbol{M}(\bar{\boldsymbol{x}} - \boldsymbol{x}) - \boldsymbol{n}'\|_{\boldsymbol{W}}^2\right] \\
&= \mathbb{E}\left[(\bar{\boldsymbol{x}} - \boldsymbol{x})^{\mathsf{T}}\boldsymbol{M}^{\mathsf{T}}\boldsymbol{P}'^{\mathsf{T}}\boldsymbol{W}\boldsymbol{P}'\boldsymbol{M}(\bar{\boldsymbol{x}} - \boldsymbol{x})\right] - \mathbb{E}\left[2(\boldsymbol{P}'\boldsymbol{M}(\bar{\boldsymbol{x}} - \boldsymbol{x}))^{\mathsf{T}}\boldsymbol{W}\boldsymbol{n}'\right] + \mathbb{E}\left[\|\boldsymbol{n}'\|_{\boldsymbol{W}}^2\right]. \\
&= \mathbb{E}\left[\|\overline{\boldsymbol{x}} - \boldsymbol{x}\|_2^2\right] + \text{constant},
\end{aligned}$$

where the first term equals to $\mathbb{E}\left[\|\overline{\boldsymbol{x}} - \boldsymbol{x}\|_2^2\right]$ due to the Proposition 2 that $\mathbb{E}\left[\boldsymbol{M}^{\mathsf{T}}\boldsymbol{P}'^{\mathsf{T}}\boldsymbol{W}\boldsymbol{P}'\boldsymbol{M}\right] = \boldsymbol{I}$ ; the second term equals to zero because $\boldsymbol{n}'$ is zero-mean and independent of $\boldsymbol{P}'$ and $\boldsymbol{x}$; The third term, $\|\boldsymbol{n}'\|_{\boldsymbol{W}}^2$, does not depend on $\boldsymbol{x}$ and contributes a constant that does not affect the optimization for training the MMSE estimator $\mathsf{R}$. $\qquad\square$

# B. Experiment Details

## B.1. Implementation details of CS-MRI tasks

**Dataset.** We simulated multi-coil subsampled measurements using T2-weighted human brain MRI data from the open-access fastMRI dataset, which comprises 4,912 fully sampled multi-coil slices for training and 470 slices for testing. Each slice has been cropped into a complex-valued image with dimensions $320 \times 320$. The coil sensitivity maps for each slice are precomputed using the ESPIRiT algorithm (Uecker et al., 2014). We simulated a Cartesian sampling pattern that subsamples along the $k_y$ dimension while fully sampling along the $k_x$ dimension.

**Subsampling pattern for CS-MRI.** In this paper, we explored two types of subsampling patterns for MRI reconstruction tasks. All undersampling masks were generated by first including a set number of *auto-calibration signal (ACS)* lines, ensuring a fully-sampled central k-space region.

Figure 4 illustrates the k-space trajectories for both random and uniform (equidistant) subsampling at acceleration factors of 4, 6, and 8. Notably, different patterns were used for training and testing. During training, our restoration prior was only trained on a uniform mask with a subsampling rate of 6. However, for inference, we employed both uniform and random masks at subsampling rates of 4 and 6, creating a mismatch between the pre-trained restoration prior and the test configurations.

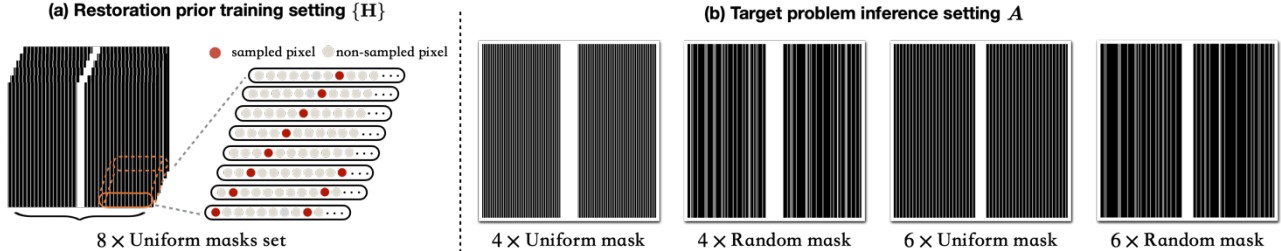

Figure 4: Illustration of the undersampling masks used for CS-MRI in this work. **(a)** The eight different $8\times$ uniform masks used for training the restoration prior. **(b)** The inference setting for ShaRP, demonstrating how the prior trained on the masks in **(a)** can be applied to solve other problems without retraining.

---

**Algorithm 2** Supervised Training of CS-MRI Restoration Network

---

**Require:** dataset: $p(\boldsymbol{x})$, sampling operator set: $\{\boldsymbol{M}_1, \boldsymbol{M}_1, \cdots, \boldsymbol{M}_1\}$, Restoration model: $\mathsf{R}_\theta(\cdot, \alpha)$

  **repeat:**
    $\boldsymbol{x} \sim p(\boldsymbol{x}), \boldsymbol{M} \sim \{\boldsymbol{M}_1, \boldsymbol{M}_2, \cdots, \boldsymbol{M}_8\}, \boldsymbol{e} \sim \mathcal{N}(0, \sigma^2\mathbf{I}), \alpha \sim \mathcal{U}([0,1])$
    $\boldsymbol{y} = \boldsymbol{M}\boldsymbol{x} + \boldsymbol{e}$
    $\min_\theta \left\| \mathsf{R}_\theta \left( (1-\alpha)\boldsymbol{x} + \alpha \boldsymbol{M}^\mathsf{T} \boldsymbol{y}; \alpha \right) - \boldsymbol{x} \right\|_2^2$
  **until converge**

---

### B.1.1. Implementation of Supervised Prior for CS-MRI

**Models training for supervised case.** We use the same U-Net architecture as employed in the official implementation of DDS[2] for $\mathsf{R}(\cdot; \alpha)$. For the supervised learning case, we select 1,000 different $\alpha$ values to train the model, following the $\alpha$ schedule outlined by I$^2$SB (Liu et al., 2023). The model is trained with Adam optimizer with a learning rate of $5 \times 10^{-5}$. As shown in Algorithm 2, we train our supervised learning model using eight different masks for $8\times$ uniform sampling CS-MRI reconstruction. In the pseudocode, $\{\boldsymbol{M}_1, \boldsymbol{M}_2, \cdots, \boldsymbol{M}_8\}$ represent the eight different MRI degradation operators, each defined by a unique sampling pattern, as shown in Figure 4 (a). This results in a total of 8,000 possible combinations of $\alpha$ values and sampling masks, effectively creating an ensemble of restoration priors during training.

**Inference with a Subset of the Ensemble (Supervised Case).** During inference, to simplify computation and focus on the most effective priors, we use only a subset of the supervised trained ensemble. Specifically, we fix the $\alpha$ value to a particular choice (e.g., $\alpha = 0.5$) and use the 8 different sampling masks $\{\boldsymbol{M}_1, \boldsymbol{M}_2, \cdots, \boldsymbol{M}_8\}$, resulting in 8 restoration priors.

---

[2]https://github.com/HJ-harry/DDS

**Step size and regularization parameter.** To ensure fairness, for each problem setting, each method—both proposed and baseline—is fine-tuned for optimal PSNR using 10 slices from a validation set separate from the test set. The same step size $\gamma$ and regularization parameter $\tau$ are then applied consistently across the entire test set.

**Baseline details.** We compare ShaRP with several variants of denoiser- and diffusion model-based methods. For denoiser-based approaches, we include PnP-FISTA (Kamilov et al., 2023), PnP-ADMM (Chan et al., 2017). PnP-FISTA and PnP-ADMM correspond to the FISTA and ADMM variants of PnP, both utilizing AWGN denoisers built on DRUNet (Zhang et al., 2022). For diffusion model-based methods, we compare with DPS (Chung et al., 2023) and DDS (Chung et al., 2024), which use pre-trained diffusion models as priors and apply different posterior sampling strategies to address general inverse problems. We use the same pre-trained diffusion model configuration as outlined in the DDS paper. For all baseline methods, we fine-tuned their parameters to maximize the PSNR value. Notably, both the DRUNet denoiser and the diffusion model were trained using the same dataset employed for training our restoration prior. For a fair comparison, the diffusion model pre-trained for DDS and DPS use the same network architecture as our restoration network . All models are trained from scratch on the fastMRI training set, following the architecture settings provided in DDS[3]. We also compared with method that also use the deep restoration prior to solve general inverse problem: DRP (Hu et al., 2024c). For DRP, we utilize the same pre-trained restoration network as in ShaRP. However, instead of employing a set of degradation priors, DRP uses a single fixed prior. For a fair comparison, we selected the optimal fixed prior—defined by a fixed $\alpha$ and subsampling mask—based on PSNR performance on the validation set, and applied it accordingly.

### B.1.2. IMPLEMENTATION OF SELF-SUPERVISED PRIOR FOR CS-MRI

---

**Algorithm 3** Self-Supervised Training of CS-MRI Restoration Network

---

**Require:** dataset: $p(\boldsymbol{y}_i, \boldsymbol{M}_i, \boldsymbol{y}_j, \boldsymbol{M}_j)$, Restoration model: $\mathsf{R}_\theta(\cdot)$
  **repeat:**
    $\boldsymbol{y}_i, \boldsymbol{M}_i, \boldsymbol{y}_j, \boldsymbol{M}_j \sim p(\boldsymbol{y}_i, \boldsymbol{M}_i, \boldsymbol{y}_j, \boldsymbol{M}_j), \boldsymbol{e} \sim \mathcal{N}(0, \sigma^2 \mathbf{I})$
    $\min_\theta \left\| \boldsymbol{M}_j \mathsf{R}_\theta \left( \boldsymbol{M}_i^\mathsf{T}(\boldsymbol{y}_i + \boldsymbol{e}) \right) - \boldsymbol{y}_j \right\|_{\boldsymbol{W}}^2$
  **until converge**

---

**Models training for (Self-Supervised Case).** For self-supervised training, the ground truth reference $\boldsymbol{x}$ is not available as a label. Instead, as shown in Algorithm 3, we work with pairs of subsampled measurements, $y_i$ and $y_j$, along with their corresponding sampling operators, $\boldsymbol{M}_i$ and $\boldsymbol{M}_j$. These paired measurements exhibit significant overlap within the shared *auto-calibration signal (ACS)* region, which increases the weighting of these overlapping k-space regions. Following the approach proposed by SSDEQ (Gan et al., 2023b), we introduce a diagonal weighting matrix $\boldsymbol{W}$ to account for the oversampled regions in the loss function. By incorporating this weighted loss, we are able to train our MMSE restoration operator using incomplete measurements alone. Furthermore, unlike the supervised case where we use the combination of $\alpha$ values to form an ensemble, in the self-supervised setting, we construct the ensemble using only eight different sampling masks across the entire dataset.

**Inference Using All Restoration Priors (Self-Supervised Case).** During inference in the self-supervised setting, we utilize all 8 restoration priors corresponding to the different sampling masks. By incorporating the entire ensemble, we fully leverage its capacity to remove the artifacts and enhance reconstruction performance.

**Step size and regularization parameter.** To ensure fairness, for each problem setting, each method—both proposed and baseline—is fine-tuned for optimal PSNR using 10 slices from a validation set separate from the test set. The same step size $\gamma$ and regularization parameter $\tau$ are then applied consistently across the entire test set.

**Baseline details.** In the self-supervised setting, we compared ShaRP with two widely used traditional methods: TV (Block et al., 2007) and GRAPPA (Griswold et al., 2002), both of which address the restoration problem without requiring fully-sampled references. Additionally, we included SPICER (Hu et al., 2024d), a recent state-of-the-art self-supervised deep unrolling method designed for MRI reconstruction using only pairs of undersampled measurements. To ensure consistency, we trained the SPICER model on the same amount of paired data used for training our restoration prior in the $8\times$ uniform CS-MRI setting and applied it to other CS-MRI configurations.

---

[3]https://github.com/HJ-harry/DDS

---

**Algorithm 4** Gaussian Deblurring Restoration network training

---

**Require:** dataset:$p(\boldsymbol{x}, \boldsymbol{y})$, Gaussian blur operator: $\boldsymbol{K}$, $\mathsf{R}_\theta(\cdot, \alpha)$

  **repeat:**

    $\boldsymbol{x} \sim p(\boldsymbol{x}), \boldsymbol{e} \sim \mathcal{N}(0, \sigma^2 \mathbf{I}), \alpha \sim \mathcal{U}([0,1])$

    $\min_\theta \|\mathsf{R}_\theta\left((1-\alpha)\boldsymbol{x} + \alpha\boldsymbol{K}\boldsymbol{x}; \alpha\right) - \boldsymbol{x}\|_2^2$

  **until converge**

---

## B.2. Implementation details of SISR tasks

**Restoration Model training.** We use the same U-Net architecture as the Gaussian deblurring model provided by I$^2$SB[4]. Utilizing the pre-trained checkpoints from their repository, we fine-tune our model accordingly. Specifically, we align with their codebase and configure the model type to OT-ODE to satisfy our MMSE restoration operator assumption.

To create an ensemble of restoration priors, we consider a family of degradation operators that are convex combinations of the identity mapping $\mathbf{I}$ and the Gaussian blur operator $\boldsymbol{K}$. The blurring operator $\boldsymbol{K}$ corresponds to convolution with a Gaussian blur kernel of size $31 \times 31$ and standard deviation 3. Specifically, we define the degradation operator as $\mathbf{H}_\alpha = (1-\alpha)\mathsf{I} + \alpha\boldsymbol{K}$, where $\alpha \in [0,1]$ controls the degradation level. By varying $\alpha$, we generate multiple degradation operators, allowing us to train the restoration network $\mathsf{R}$ to handle all these operators, expressed as $\mathsf{R}(\boldsymbol{s}, \mathbf{H}_\alpha) = \mathbb{E}[\boldsymbol{x}|\boldsymbol{s}, \mathbf{H}_\alpha]$, where $\boldsymbol{s}$ is the degraded image and $\boldsymbol{x}$ is the original image.

We select 1,000 different $\alpha$ values from the interval $[0,1]$, following the $\alpha$ schedule outlined by I$^2$SB (Liu et al., 2023). This results in 1,000 different degradation operators $\mathbf{H}_\alpha$, effectively creating an ensemble of restoration priors during training. The model is trained using the Adam optimizer with a learning rate of $5 \times 10^{-5}$.

**Inference with a Subset of the Ensemble.** During inference, to simplify computation and focus on the most effective priors, we use only a subset of the supervised trained ensemble. Specifically, we select 6 $\alpha$ values, resulting in 6 restoration priors.

**Step size and regularization parameter.** To ensure fairness, for each problem setting, each method—both proposed and baseline—is fine-tuned for optimal PSNR using 5 images from a validation set separate from the test set. The same step size $\gamma$ and regularization parameter $\tau$ are then applied consistently across the entire test set.

**Baseline details.** We compare ShaRP against several denoiser- and diffusion model-based methods. For denoiser-based approaches, we evaluate DPIR (Zhang et al., 2022), which relies on half-quadratic splitting (HQS) iterations with DRUNet denoisers. For diffusion model-based methods, we compare with DPS (Chung et al., 2023), DDNM (Wang et al., 2023), and DiffPIR (Zhu et al., 2023). These methods all use the same pre-trained diffusion models as priors, but each employs a distinct posterior sampling strategy to solve general inverse problems. We specifically use the pre-trained diffusion model from DiffPIR. We also compare our approach with DRP (Hu et al., 2024c), a method that leverages deep restoration priors for solving general inverse problems. For a fair comparison, we use the same pre-trained deblurring network as in ShaRP for DRP. However, instead of employing a set of degradation priors, DRP uses a single fixed prior. For a fair comparison, we selected the optimal fixed prior—defined by a fixed $\alpha$ based on PSNR performance on the validation set, and applied it accordingly. For all baselines, we fine-tuned their parameters to maximize PSNR performance. Notably, the diffusion model backbone for all diffusion-based baselines was trained on the same dataset used to train our restoration prior.

---

[4]https://github.com/NVlabs/I2SB

## C. Additional results for CS-MRI

### C.1. Performance of ShaRP for uniform and random subsampling setting

Due to space constraints, we present only the quantitative performance for the $4\times$ uniform subsampling setting in the main paper. In this section, we further evaluate ShaRP's performance on both uniform random subsampling setting, with two sub-sampling rates ($4\times$ and $6\times$), and three noise levels ($\sigma = 0.005$, $0.01$, and $0.015$).

| | $4\times$ Uniform | | | | | | $6\times$ Uniform | | | | | |
|---|---|---|---|---|---|---|---|---|---|---|---|---|
| Noise level | $\sigma = 0.005$ | | $\sigma = 0.010$ | | $\sigma = 0.015$ | | $\sigma = 0.005$ | | $\sigma = 0.010$ | | $\sigma = 0.015$ | |
| Metrics | PSNR | SSIM | PSNR | SSIM | PSNR | SSIM | PSNR | SSIM | PSNR | SSIM | PSNR | SSIM |
| Zero-filled | 26.93 | 0.848 | 26.92 | 0.847 | 26.90 | 0.848 | 22.62 | 0.728 | 22.60 | 0.726 | 22.59 | 0.721 |
| TV | 31.17 | 0.923 | 31.08 | 0.921 | 30.91 | 0.915 | 25.00 | 0.806 | 24.94 | 0.803 | 24.33 | 0.755 |
| PnP-FISTA | 35.88 | 0.938 | 31.14 | 0.894 | 30.32 | 0.846 | 26.30 | 0.822 | 25.97 | 0.786 | 25.46 | 0.747 |
| PnP-ADMM | 35.76 | 0.941 | 32.36 | 0.878 | 30.66 | 0.838 | 26.13 | 0.808 | 25.90 | 0.776 | 25.51 | 0.742 |
| DRP | 35.52 | 0.936 | 32.32 | 0.914 | 30.57 | 0.901 | 29.51 | 0.872 | 28.52 | 0.882 | 28.35 | 0.876 |
| DPS | 32.62 | 0.888 | 31.39 | 0.870 | 30.29 | 0.856 | 30.53 | 0.862 | 29.41 | 0.843 | 28.63 | 0.830 |
| DDS | 35.21 | 0.937 | 35.03 | 0.935 | 34.51 | 0.925 | 31.02 | 0.889 | 30.84 | 0.888 | 30.79 | 0.888 |
| ShaRP | **37.59** | **0.963** | **35.81** | **0.951** | **34.92** | **0.942** | **33.42** | **0.940** | **32.86** | **0.932** | **32.09** | **0.922** |

Table 5: Quantitative comparison of ShaRP with several baselines for CS-MRI using uniform masks at undersampling rates of 4 and 6 on fastMRI dataset. The **best** and second best results are highlighted. Notably, ShaRP outperforms SOTA methods based on denoisers and diffusion models.

| | $4\times$ Random | | | | | | $6\times$ Random | | | | | |
|---|---|---|---|---|---|---|---|---|---|---|---|---|
| Noise level | $\sigma = 0.005$ | | $\sigma = 0.010$ | | $\sigma = 0.015$ | | $\sigma = 0.005$ | | $\sigma = 0.010$ | | $\sigma = 0.015$ | |
| Metrics | PSNR | SSIM | PSNR | SSIM | PSNR | SSIM | PSNR | SSIM | PSNR | SSIM | PSNR | SSIM |
| Zero-filled | 25.83 | 0.815 | 25.81 | 0.812 | 25.76 | 0.807 | 22.68 | 0.724 | 22.67 | 0.722 | 22.67 | 0.719 |
| TV | 28.14 | 0.866 | 28.06 | 0.863 | 27.96 | 0.859 | 24.55 | 0.782 | 24.33 | 0.750 | 24.28 | 0.736 |
| PnP-FISTA | 29.31 | 0.863 | 28.40 | 0.817 | 27.49 | 0.799 | 26.01 | 0.797 | 25.63 | 0.756 | 24.94 | 0.717 |
| PnP-ADMM | 28.83 | 0.842 | 28.39 | 0.816 | 27.70 | 0.786 | 25.59 | 0.776 | 25.19 | 0.740 | 24.93 | 0.728 |
| DRP | 29.97 | 0.880 | 29.37 | 0.839 | 28.31 | 0.794 | 26.98 | 0.866 | 26.78 | 0.853 | 26.49 | 0.821 |
| DPS | 31.72 | 0.874 | 30.45 | 0.857 | 29.50 | 0.843 | 30.32 | 0.856 | 29.36 | 0.824 | 27.99 | 0.810 |
| DDS | 32.41 | 0.910 | 32.37 | 0.906 | 32.25 | 0.901 | 30.59 | 0.876 | 30.35 | 0.874 | 30.31 | 0.879 |
| ShaRP | **34.66** | **0.949** | **33.57** | **0.920** | **33.18** | **0.931** | **31.53** | **0.924** | **31.46** | **0.918** | **31.45** | **0.914** |

Table 6: Quantitative comparison of ShaRP with several baselines for CS-MRI using random masks at undersampling rates of 4 and 6 on fastMRI dataset. The **best** and second best results are highlighted. Notably, ShaRP outperforms SOTA methods based on denoisers and diffusion models.

Table 6 provides a quantitative comparison of reconstruction performance across different acceleration factors and noise levels using a uniform sub-sampling mask. In all configurations, ShaRP consistently outperforms the baseline methods. The use of a set of restoration operators clearly enhances ShaRP's performance, highlighting the effectiveness of employing multiple operators to maximize the regularization information provided by the restoration model. Figure 6 presents visual reconstructions for two test scenarios, where ShaRP accurately recovers fine brain details, particularly in the zoomed-in regions, while baseline methods tend to oversmooth or introduce artifacts. These results highlight ShaRP's superior ability to manage structured artifacts and preserve fine details, outperforming both denoiser-based and diffusion model-based methods.

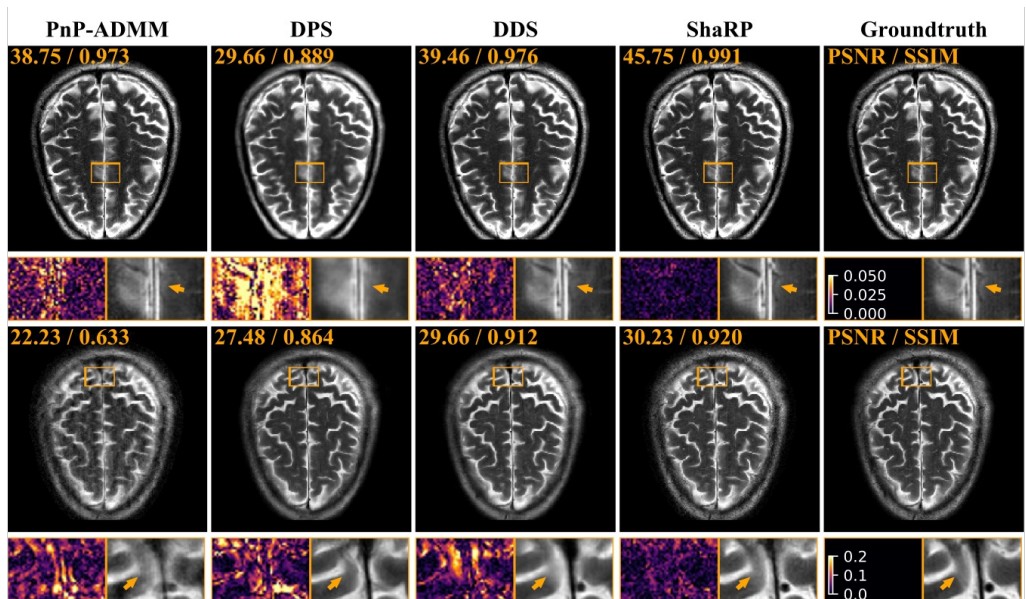

Figure 5: Visual comparison of ShaRP with baseline methods on CS-MRI. The top row shows results for a $4\times$ random mask with noise $\sigma = 0.005$, and the bottom row for a $6\times$ random mask with noise $\sigma = 0.015$. PSNR and SSIM values are in the top-left corner of each image. Error maps and zoomed-in areas highlight differences. Notably, ShaRP with stochastic priors outperforms state-of-the-art methods using denoiser and diffusion model priors.

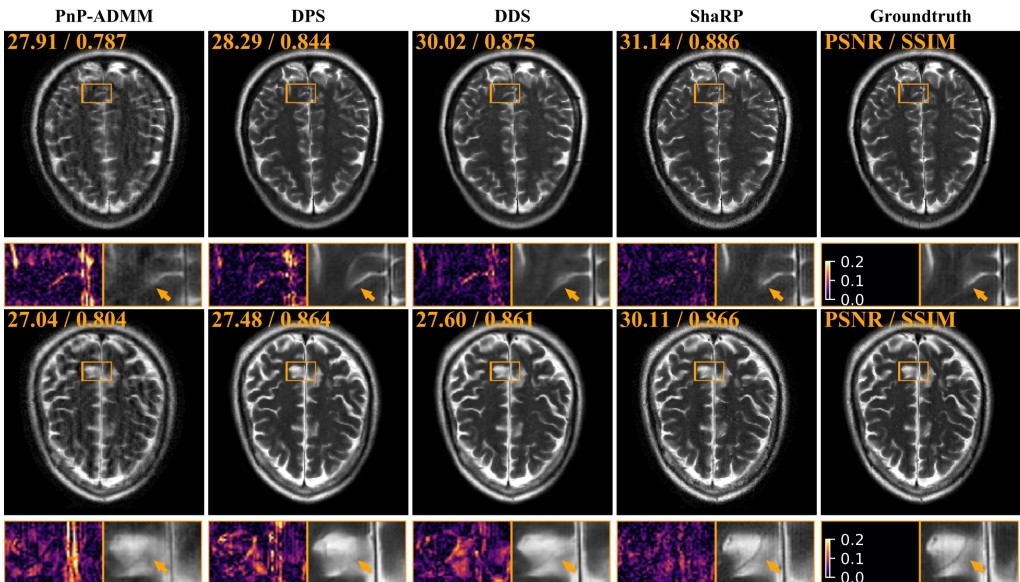

Figure 6: Visual comparison of ShaRP with baseline methods on CS-MRI for $6\times$ random sampling mask with noise $\sigma = 0.015$. PSNR and SSIM values are in the top-left corner of each image. Error maps and zoomed-in areas highlight differences. Notably, ShaRP with stochastic priors outperforms state-of-the-art methods using denoiser and diffusion model priors.

## C.2. Performance of ShaRP with Self-supervised restoration priors

| | 4× Random | | | | | 6× Random | | | | | |
|---|---|---|---|---|---|---|---|---|---|---|---|---|
| Noise level | $\sigma = 0.005$ | | $\sigma = 0.010$ | | $\sigma = 0.015$ | | $\sigma = 0.005$ | | $\sigma = 0.010$ | | $\sigma = 0.015$ | |
| Metrics | PSNR | SSIM | PSNR | SSIM | PSNR | SSIM | PSNR | SSIM | PSNR | SSIM | PSNR | SSIM |
| PnP-ADMM | 28.83 | 0.842 | 28.39 | 0.816 | 27.70 | 0.786 | 25.59 | 0.776 | 25.19 | 0.740 | 24.93 | 0.728 |
| ADMM-TV | 28.14 | 0.866 | 28.06 | 0.863 | 27.96 | 0.859 | 24.55 | 0.782 | 24.33 | 0.750 | 24.28 | 0.736 |
| GRAPPA | 28.09 | 0.792 | 25.39 | 0.699 | 23.94 | 0.649 | 25.67 | 0.737 | 23.72 | 0.646 | 22.51 | 0.595 |
| SPICER | 31.87 | 0.901 | 31.67 | 0.889 | 31.50 | 0.887 | 30.18 | 0.871 | 30.05 | 0.863 | 30.01 | 0.860 |
| ShaRP$^{self}$ | **33.87** | **0.909** | **33.64** | **0.900** | **33.21** | **0.892** | **30.87** | **0.899** | **30.36** | **0.890** | **30.21** | **0.875** |

Table 7: PSNR (dB) and SSIM values for ShaRP with a self-supervised pre-trained restoration operator, compared to several baselines for CS-MRI with random undersampling masks at rates of 4 and 6 on the fastMRI dataset. The **best** and second best results are highlighted. For reference, the highlighted row presents a PnP method using a Gaussian denoiser, which requires fully sampled data for training. Note the excellent performance of ShaRP even using priors trained without fully-sampled ground-truth data.

## C.3. Convergence Performance of ShaRP with supervised and self-supervised priors

Figure 7 illustrates the convergence behavior of ShaRP with both supervised and self-supervised learned restoration priors on the test set with an acceleration factor of $R = 6$ and additional noise $\sigma = 0.01$. As observed, ShaRP converges stably under both settings. However, the self-supervised prior exhibits a performance gap compared to the supervised one, indicating that it does not serve as a perfect MMSE estimator in practice. Despite this, it still enables stable convergence as an ensemble of priors for ShaRP, which aligns with our convergence analysis for inexact MMSE estimators presented in Proposition 1.

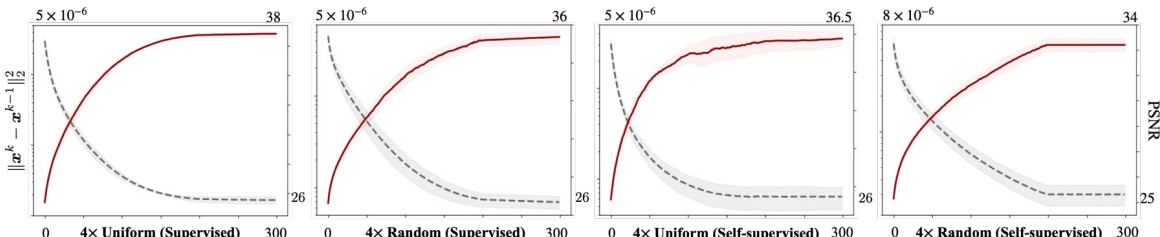

Figure 7: Convergence of ShaRP for $4\times$ accelerated MRI reconstruction on the fastMRI dataset. **(a)-(b)** depict the convergence behavior of ShaRP using restoration operators trained in a supervised manner, while **(c)-(d)** correspond to those trained in a self-supervised manner. The plots illustrate the average distance $\|\boldsymbol{x}^k - \boldsymbol{x}^{k-1}\|_2^2$ and PSNR relative to the ground truth, as a function of the iteration number, with shaded regions representing the standard deviation. Note the stable convergence of ShaRP with both types of priors.

## C.4. Performance of additional baseline methods on matched and mismatched settings

In this section, we highlight an important observation: pre-trained restoration networks typically exhibit poor generalization to mismatched settings. We chose two commonly used methods (SwinIR (Liang et al., 2021) and E2E-VarNet (Sriram et al., 2020)) for the specific setting of CS-MRI. We trained them on the same $8\times$ uniform subsampling setting as our restoration prior and directly applied them to solve both matched and mismatched problems, as ShaRP did. As shown in the Table 8, the baseline method's performance drops significantly under mismatched conditions, whereas ShaRP maintains stable performance and convergence guarantees. This demonstrates ShaRP's ability to adapt pre-trained restoration models as priors and use it to solve problems under mismatched settings. As shown in the Figure 8, due to the mismatched settings, the two baseline methods suffer from over-smoothing, lack important details, and exhibit artifacts, whereas ShaRP still provides high-quality reconstruction performance. This indicates that ShaRP can balance data fidelity and the artifact removal capabilities of the prior model, leading to an artifact-free reconstruction that preserves important details.

| Settings | $4\times$ Uniform | | $4\times$ Random | | $6\times$ Uniform | | $6\times$ Random | | $8\times$ Uniform | |
|---|---|---|---|---|---|---|---|---|---|---|
| Metrics | PSNR | SSIM | PSNR | SSIM | PSNR | SSIM | PSNR | SSIM | PSNR | SSIM |
| SwinIR | 24.78 | 0.849 | 25.09 | 0.841 | 29.55 | 0.907 | 27.98 | 0.819 | 29.37 | 0.898 |
| E2E-VarNet | 35.40 | 0.957 | 33.48 | 0.945 | 32.79 | 0.936 | 31.02 | 0.913 | 32.59 | 0.919 |
| ShaRP | 37.59 | 0.963 | 34.66 | 0.949 | 33.42 | 0.940 | 31.53 | 0.924 | 32.37 | 0.907 |

Table 8: Quantitative comparison of ShaRP with task-specific baselines trained on the $8\times$ uniform mask. Baselines perform well in matched settings (highlighted in the table) but show a significant drop under mismatched conditions. In contrast, ShaRP remains robust, handling both matched and mismatched scenarios effectively.

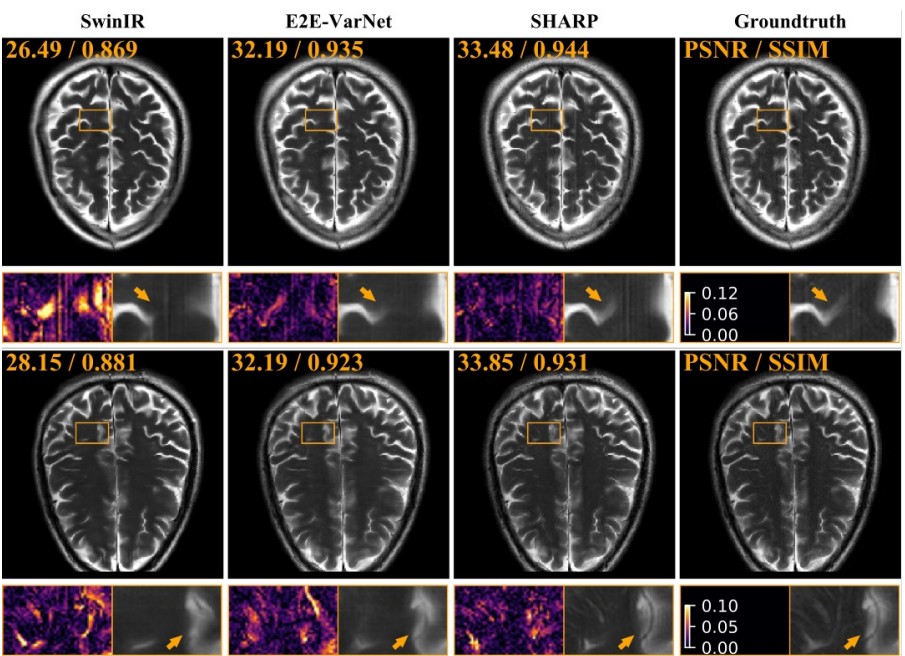

Figure 8: Visual comparison of ShaRP with task-specific baseline methods on CS-MRI for $6\times$ random sampling mask with noise $\sigma = 0.015$. PSNR and SSIM values are in the top-left corner of each image. Error maps and zoomed-in areas highlight differences. Notably, ShaRP with stochastic priors outperforms state-of-the-art methods using denoiser and diffusion model priors.

## C.5. Additional Evaluation: Non-Cartesian CS-MRI Generalization

In this section, we present an additional evaluation designed to further assess ShaRP's generalization capabilities. This new experimental setting explores non-Cartesian sampling for Compressed Sensing MRI (CS-MRI), specifically employing a 2D

Gaussian sampling mask. A key aspect of this evaluation is that the restoration network, originally trained for $8\times$ uniform Cartesian sampling, was directly applied to this unseen sampling scheme without any retraining or fine-tuning. The results, summarized in Table 9, provide compelling evidence of ShaRP's robustness to diverse sampling patterns and highlight its strong generalization performance.

Table 9: Performance comparison on CS-MRI with a 2D Gaussian sampling mask. The restoration network was trained on $8\times$ uniform Cartesian sampling and applied directly without retraining or adaptation.

| Method | PSNR | SSIM |
|---|---|---|
| PnP-ADMM | 31.33 | 0.917 |
| DPS | 32.01 | 0.904 |
| DDS | 33.19 | 0.924 |
| DRP | 32.70 | 0.921 |
| ShaRP | **34.01** | **0.942** |

## D. Additional visual results for SISR

In this section, we present additional visual results to numerical comparisons for the SISR task.

### D.1. Additional visual results against baselines

As illustrated in Figure 9 and Figure 10, ShaRP outperforms all baseline approaches under both blur kernel settings, achieving higher PSNR and SSIM values. Moreover, we maintain superior data consistency with the measurements while achieving enhanced perceptual quality. The use of an ensemble of deblurring priors enables our method to recover fine details at varying corruption levels, contributing to the improved performance.

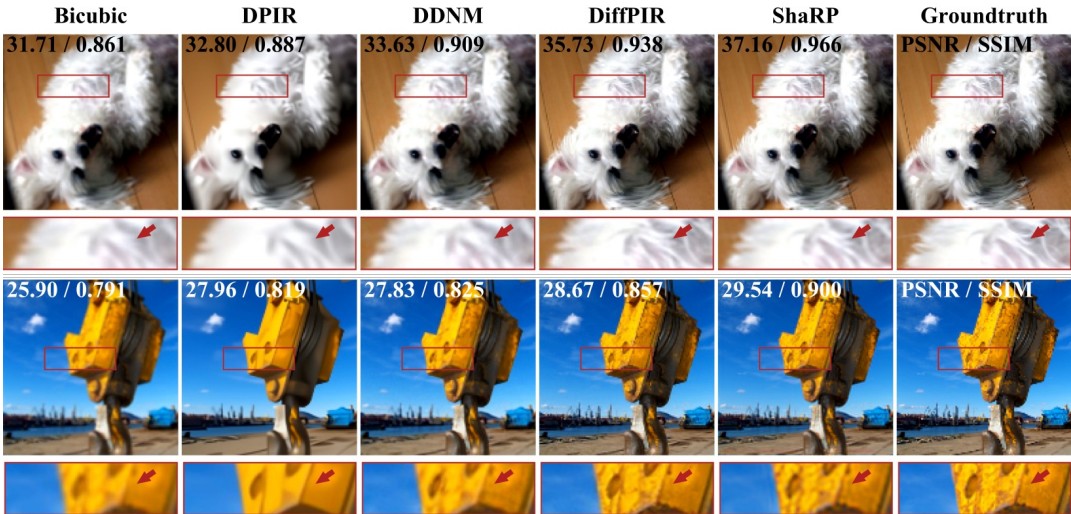

Figure 9: Visual comparison of ShaRP with several well-known methods on $2\times$ SISR with gaussian blur kernel with $\sigma = 1.5$. The quantities in the top-left corner of each image provide PSNR and SSIM values for each method. The squares at the bottom of each image visualize the zoomed area in the image.

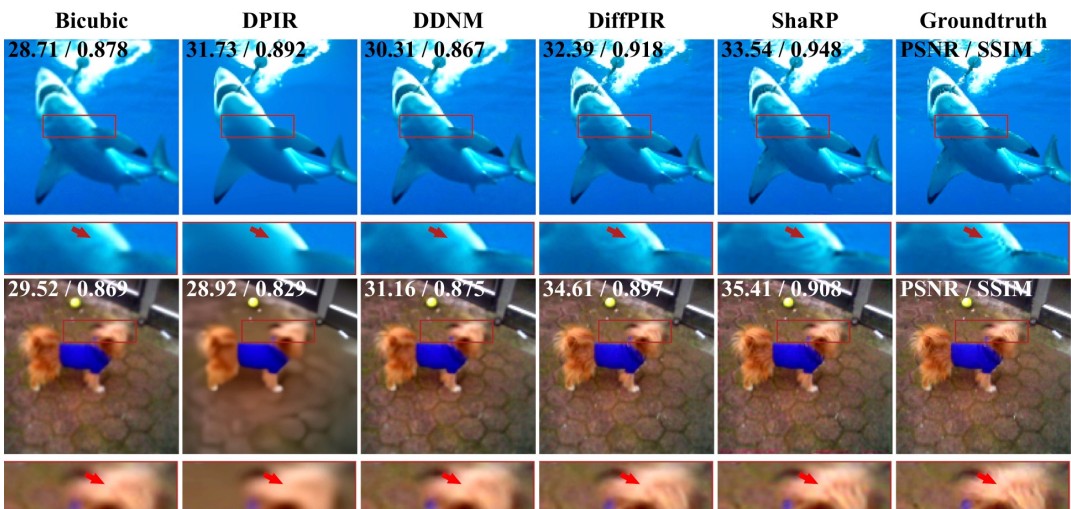

Figure 10: Visual comparison of ShaRP with several well-known methods on $2\times$ SISR with gaussian blur kernel with $\sigma = 1.25$. The quantities in the top-left corner of each image provide PSNR and SSIM values for each method. The squares at the bottom of each image visualize the zoomed area in the image.

### D.2. Additional visual results against DRP

To further emphasize the necessity and advantages of using an ensemble of deblurring priors, as opposed to a fixed prior like in DRP (Hu et al., 2024c), we provide additional visual comparison results. As shown in Figure 11, ShaRP consistently recovers finer details, resulting in improved PSNR and SSIM scores, along with enhanced perceptual performance.

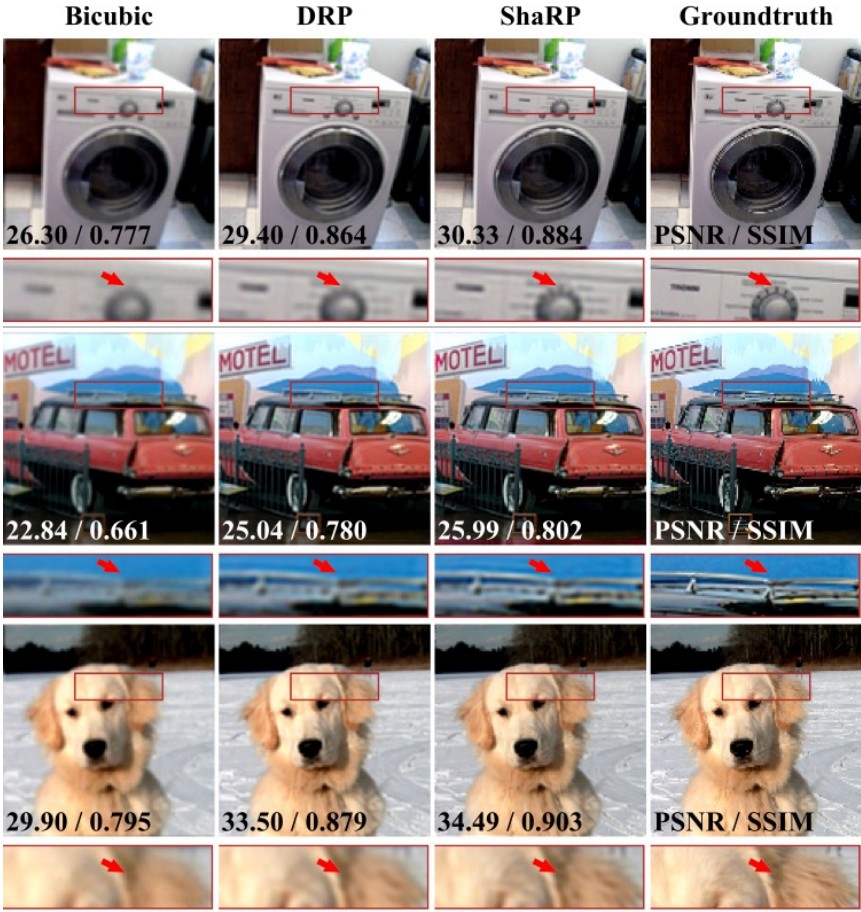

Figure 11: Visual comparison of ShaRP with DRP on $2\times$ SISR with gaussian blur kernel with $\sigma = 1.5$. The quantities in the bottom-left corner of each image provide PSNR and SSIM values for each method. The squares at the bottom of each image visualize the zoomed area in the image.

### D.3. Additional Comparison against DDRM and DiffIR

To further evaluate ShaRP's performance against state-of-the-art diffusion-based methods, we included two additional baselines for comparison: DDRM (Kawar et al., 2022) and DiffIR (Xia et al., 2023). The experiment setting is $2\times$ SISR task with gaussian blur kernel with $\sigma = 1.25$ on ImageNet dataset. For DDRM, we utilized the same pre-trained unconditional diffusion backbone as DiffPIR, DDNM, and DDS, but followed the sampling procedure outlined in their original paper. For DiffIR, we directly used the provided checkpoint from the authors.

| Metrics | PSNR | SSIM | LPIPS |
|---------|------|------|-------|
| DPIR | 28.10 | 0.809 | 0.305 |
| DDNM | 27.53 | 0.786 | 0.240 |
| DPS | 24.68 | 0.661 | 0.395 |
| DiffPIR | 28.92 | 0.852 | **0.152** |
| DiffIR | 25.79 | 0.812 | 0.180 |
| DDRM | 28.20 | 0.845 | 0.161 |
| DRP | 29.28 | 0.868 | 0.207 |
| ShaRP | **30.09** | **0.891** | 0.179 |

Table 10: Quantitative comparison of ShaRP with several additional baselines for $2\times$ SISR with gaussian blur kernel with $\sigma = 1.25$ on ImageNet dataset. The **best** and second best results are highlighted. Notably, ShaRP outperforms SOTA methods based on denoisers and diffusion models.

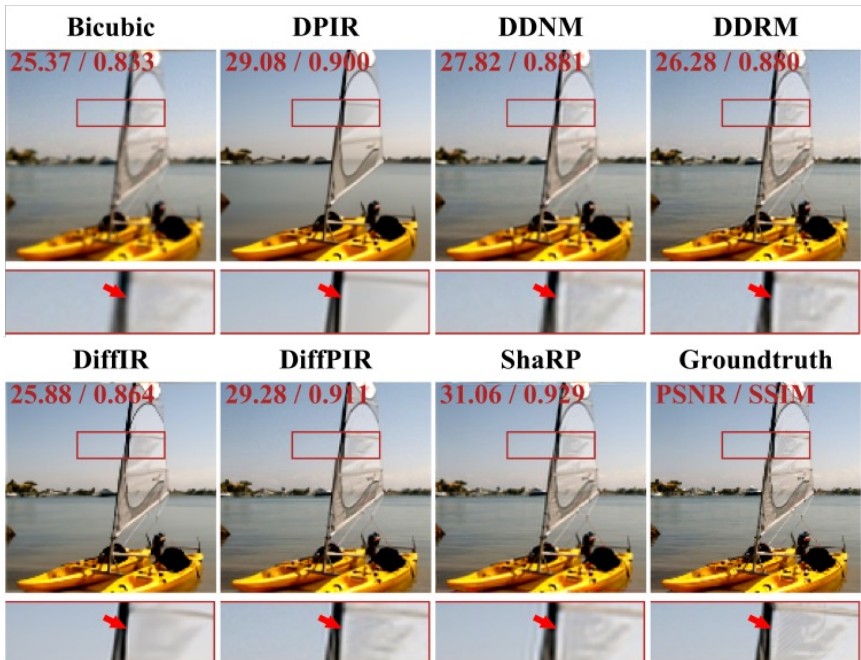

Figure 12: Visual comparison of ShaRP with additional baselines on $2\times$ SISR with gaussian blur kernel with $\sigma = 1.25$. The quantities in the bottom-left corner of each image provide PSNR and SSIM values for each method. The squares at the bottom of each image visualize the zoomed area in the image.

**D.4. Additional Evaluation: SISR with New Gaussian Kernel**

To further investigate ShaRP's robustness, we ran an additional Super-Resolution (SISR) experiment using a new Gaussian blur kernel with $\sigma = 1.0$. This test aimed to evaluate if ShaRP can maintain consistently good performance, comparable to its results with the other two kernels detailed in the main manuscript, even with this previously unseen degradation. As shown in Table 11, ShaRP continues to provide excellent performance, suggesting its strong generalization capabilities and potential effectiveness across a wider range of Gaussian blur levels.

Table 11: SISR performance with a new Gaussian blur kernel ($\sigma = 1.0$).

| Method | PSNR (dB) | SSIM | LPIPS ↓ | FID ↓ |
|---|---|---|---|---|
| DPIR | 28.45 | 0.854 | 0.247 | 82.90 |
| DDRM | 27.26 | 0.803 | 0.209 | 44.77 |
| DiffPIR | 28.37 | 0.841 | 0.215 | 40.59 |
| DRP | 28.43 | 0.853 | 0.236 | 75.29 |
| ShaRP | **28.70** | **0.858** | **0.226** | **69.75** |

# E. Additional Experiments

In this section, we include two additional ablation studies to further highlights ShaRP's capability to leverage restoration priors for solving general inverse problems, as well as to evaluate its performance under different hyperparameter settings.

## E.1. Ablation study on using SR prior for CS-MRI task

To demonstrate the flexibility of our approach in integrating diverse restoration models to address general inverse problems, we conducted an additional ablation study using a pre-trained super-resolution network as a prior for solving the CS-MRI problem.

---

**Algorithm 5** MRI Super Resolution network training

---

**Require:** dataset:$p(\boldsymbol{x}, \boldsymbol{y})$, $4\times$ bicubic downsampling operator: $\boldsymbol{K}$, $\mathsf{R}_\theta(\cdot, \alpha)$
  **repeat:**
    $\boldsymbol{x} \sim p(\boldsymbol{x})$, $\boldsymbol{e} \sim \mathcal{N}(0, \sigma^2 \mathbf{I})$, $\alpha \sim \mathcal{U}([0, 1])$
    $\min_\theta \left\| \mathsf{R}_\theta \left( (1 - \alpha)\boldsymbol{x} + \alpha \boldsymbol{D}^\mathsf{T} \boldsymbol{D}\boldsymbol{x}; \alpha \right) - \boldsymbol{x} \right\|_2^2$
  **until converge**

---

**Models training for MRI-SR** We use the same U-Net architecture as employed in the official implementation of DDS[5] for $\mathsf{R}(\cdot; \alpha)$. To create an ensemble of restoration priors, we consider a family of degradation operators that are convex combinations of the identity mapping $\mathbf{I}$ and the Gaussian blur operator $\boldsymbol{D}$. The $4\times$ bicubic downsampling operator $\boldsymbol{D}$ corresponds to bicubic downsample with factor equals to 4. Specifically, we define the degradation operator as $\mathbf{H}_\alpha = (1 - \alpha)\mathsf{I} + \alpha \boldsymbol{D}^\mathsf{T} \boldsymbol{D}$, where $\alpha \in [0, 1]$ controls the degradation level. By varying $\alpha$, we generate multiple degradation operators, allowing us to train the restoration network R to handle all these operators, expressed as $\mathsf{R}(\boldsymbol{s}, \mathbf{H}_\alpha) = \mathbb{E}[\boldsymbol{x}|\boldsymbol{s}, \mathbf{H}_\alpha]$, where $\boldsymbol{s}$ is the degraded image and $\boldsymbol{x}$ is the original image.

We select 1,000 different $\alpha$ values from the interval $[0, 1]$, following the $\alpha$ schedule outlined by $\mathrm{I}^2\mathrm{SB}$ (Liu et al., 2023). This results in 1,000 different degradation operators $\mathbf{H}_\alpha$, effectively creating an ensemble of restoration priors during training. The model is trained using the Adam optimizer with a learning rate of $5 \times 10^{-5}$.

**Using MRI-SR model as prior for CS-MRI task.** During inference, to simplify computation and focus on the most effective priors, we use only a subset of the ensemble. Specifically, we select 6 $\alpha$ values, resulting in 6 restoration priors.

As shown in Table 12, under the $4\times$ uniform mask setting, employing the pre-trained MRI-SR model as prior allows ShaRP to outperform denoiser- and diffusion-based approaches. However, its performance remains inferior to ShaRP with a mismatched CS-MRI-specific prior. In the $4\times$ random mask setting, ShaRP with the pre-trained MRI-SR model as prior continues to surpass PnP-based methods that utilize a denoiser prior but performs worse than approaches based on diffusion models. Notably, ShaRP with a mismatched CS-MRI-specific prior consistently delivers the best performance.

| Tasks | Metrics | PnP-FISTA | PnP-ADMM | DPS | DDS | ShaRP$_{\mathrm{CS}}$ | ShaRP$_{\mathrm{SR}}$ |
|---|---|---|---|---|---|---|---|
| 4x Uniform | PSNR | 35.88 | 35.76 | 32.62 | 35.21 | **37.59** | 35.91 |
| | SSIM | 0.938 | 0.941 | 0.888 | 0.937 | **0.961** | 0.943 |
| 4x Random | PSNR | 29.31 | 28.83 | 31.72 | 32.41 | **34.66** | 30.91 |
| | SSIM | 0.863 | 0.842 | 0.874 | 0.910 | **0.949** | 0.905 |

Table 12: Quantitative comparison of ShaRP against baselines for CS-MRI reconstruction using $8\times$ CS-MRI and $4\times$ super-resolution priors, evaluated on the fastMRI dataset. Results are reported for both uniform and random undersampling masks at a 4x undersampling rate. The **best** and second best results are highlighted.

---

[5]https://github.com/HJ-harry/DDS

### E.2. Ablation Study on the number of restoration priors included in the ensemble

To evaluate the impact of the number of restoration priors, $b$, on ShaRP's performance, we conducted an ablation study in the $4\times$ CS-MRI setting with random sampling masks. We specifically examined how varying $b$ influenced reconstruction performance, offering valuable insights into ShaRP's sensitivity to this parameter and its role in achieving optimal results.

As illustrated in Figure 13 and Figure 14, increasing $b$, which corresponds to incorporating more restoration priors in the ensemble, generally enhances ShaRP's reconstruction performance.

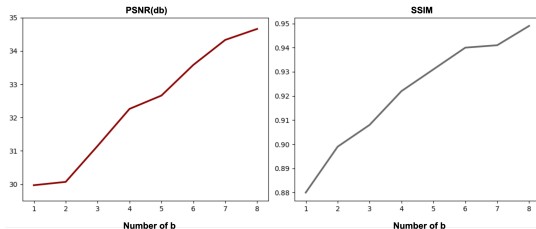

Figure 13: Performance comparison of ShaRP's CS-MRI reconstruction at $4\times$ acceleration with varying numbers of restoration priors, $b$. Left: PSNR vs. $b$; Right: SSIM vs. $b$. ShaRP with $b = 8$ consistently achieves superior results, highlighting the performance improvements gained by incorporating more restoration priors into ShaRP.

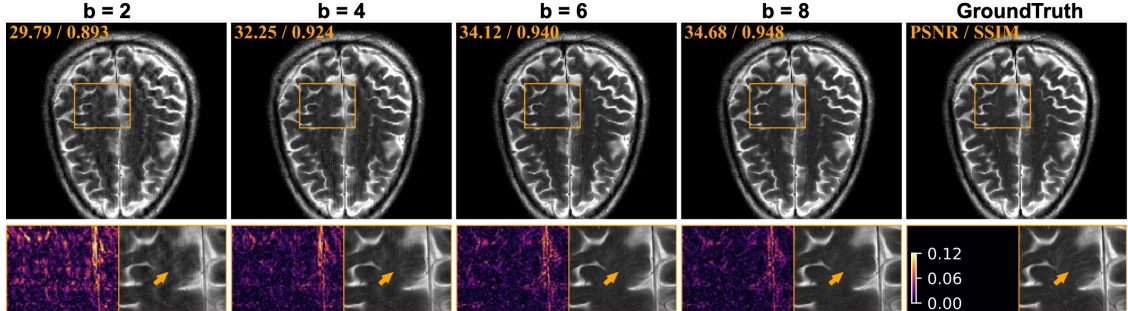

Figure 14: Visual comparison of ShaRP with varying amounts of restoration priors, denoted by $b$, in the ensemble. The PSNR and SSIM values for each method are shown in the top-left corner of each image. Zoomed-in regions, highlighted as squares at the bottom of each image, provide a closer look at key details. Notably, increasing the number of restoration priors in the ensemble enhances visual performance by effectively reducing artifacts and capturing finer details.

### E.3. Ablation Study on the Influence of Hyperparameter $\alpha$

To evaluate the impact of the hyperparameters $\alpha$ as introduced in Section B.1, we conducted an ablation study in the $4\times$ CS-MRI setting using random sampling masks, where $\alpha$ governs the selection of a specific restoration prior. Specifically, we analyzed how varying the values of $\alpha$ influenced reconstruction performance. As shown in Figure 15 demonstrates the influence of $\alpha$ on performance. A very small $\alpha$ fails to provide sufficient regularization to constrain the solution, while an excessively large $\alpha$ overly restricts the model, leading to a decline in performance. These findings highlight the importance of appropriately tuning $\alpha$ and $b$ to balance flexibility and regularization for optimal results.

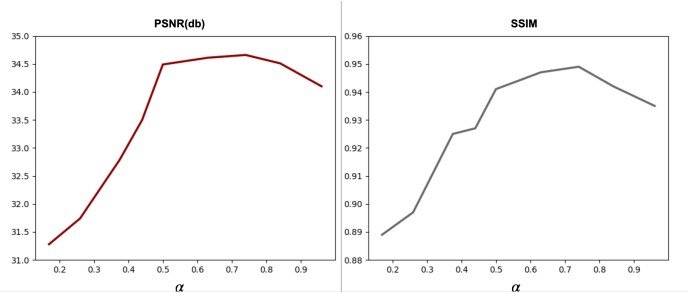

Figure 15: Performance comparison of ShaRP's CS-MRI reconstruction at $4\times$ acceleration with varying $\alpha$. Left: PSNR vs. $\alpha$; Right: SSIM vs. $\alpha$.

