# OpenReview forum: "Stochastic Deep Restoration Priors for Imaging Inverse Problems"
_ICML.cc/2025/Conference — ICML 2025 poster_

### Official Review · Reviewer_w5bu · 2025-02-20

**Overall Recommendation:** 3

**Summary:**

This paper introduces stochastic deep restoration priors, a framework leveraging an ensemble of pre-trained restoration models as priors for solving imaging inverse problems. The authors claim they minimizes a regularizer based on degraded observation likelihoods, generalizing denoiser-based methods like RED and SNORE. Experiments on MRI reconstruction and super-resolution demonstrate SOTA performance. The main contributions include theoretical convergence guarantees, adaptation to self-supervised training without fully sampled data, and improved handling of structured artifacts.

**Claims And Evidence:**

The authors claim that restoration priors outperform Gaussian denoisers is supported by experiments showing higher PSNR/SSIM on MRI and SISR tasks. Theoretical justification links ShaRP to a sound regularizer, and convergence analysis under idealized assumptions provides a partial foundation. Empirical gains over diffusion models are clear.

**Essential References Not Discussed:**

The two papers shown below handle similar tasks:
[1] Osmosis: Rgbd diffusion prior for underwater image restoration. ECCV, 2024.
[2] DreamClean: Restoring Clean Image Using Deep Diffusion Prior. ICLR, 2024.

**Experimental Designs Or Analyses:**

1. MRI experiments are thorough, but no ablation study quantifies how ShaRP’s gains scale with ensemble size b.
2. SISR uses only two Gaussian blurs, which is not enough.
3. Perceptual quality requires human evaluation.

**Methods And Evaluation Criteria:**

ShaRP builds on PnP/RED but modifies via restoration operator ensembles. Evaluation uses standard datasets, i.e., fastMRI and ImageNet. However, for MRI, validation on uniform/random sampling subsets is reasonable, but real-world non-Cartesian sampling is not tested. SISR results focus on synthetic blur kernels, but robustness to natural distortions (e.g., motion blur) is unclear.

**Other Comments Or Suggestions:**

1. Technical terms (e.g., "restoration operator") are inconsistently defined
2. Supplement should provide experiments with other degradation types..

**Other Strengths And Weaknesses:**

Strengths:
1. Theoretically grounded regularization using MMSE restoration ensembles.
2. Practical advantages in self-supervised MRI where prior denoiser training requires full data.
3. Clear empirical gains over denoiser-/diffusion-based baselines.

Weaknesses:
1. Limited diversity in tested degradations.
2. Computational costs of stochastic restoration steps are not analyzed.

**Questions For Authors:**

1. How does ShaRP’s performance/complexity scale with ensemble size?
2. Could ShaRP adapt to settings without even partially sampled data?
3. How did inexact restoration operators impact convergence in practice? Does the bias term remain stable?

**Relation To Broader Scientific Literature:**

ShaRP aligns with recent trends in learning-based priors, e.g., denoisers and diffusion, but extends them via restoration ensembles. Comparisons to PnP/RED and diffusion methods (DPS, DDS) are suitable.

**Theoretical Claims:**

In Appendix, Theorem 1 is correct, and Theorem 2 relies on ideal assumptions: Lipschitz continuity and bounded bias. While practical convergence is shown empirically, the gap between theory and practice needs discussion.

---

> ### Author Rebuttal · Authors · 2025-04-01
>
> We thank the reviewer for their valuable feedback.
>
> >1. *For concerns in Methods And Evaluation Criteria:*
>
> Prompted by your comment, we ran additional experiments that we will include in the supplementary material of the paper. The new setting considers non-Cartesian sampling for CS-MRI, using a 2D Gaussian sampling mask. Importantly, the restoration network trained for the 8× uniform Cartesian sampling was applied directly without retraining or adaptation. Our results, summarized below, provide additional evidence of ShaRP's robustness to diverse sampling patterns, highlighting its strong generalization ability.
>
> | Method     | PSNR  | SSIM  |
> |------------|-------|-------|
> | PnP-ADMM   | 31.33 | 0.917 |
> | DPS        | 32.01 | 0.904 |
> | DDS        | 33.19 | 0.924 |
> | DRP        | 32.70 | 0.921 |
> | ShaRP      | 34.01 | 0.942 |
>
> It would certainly be interesting to explore additional natural distortions in SISR tasks, such as motion blur, in future work. We will mention this revised manuscript.
>
> >2. *For concerns in Theoretical Claims:*
>
> Thanks for your suggestion. We will include an expanded discussion about the two technical assumptions. Lipschitz continuity is a standard and widely-used assumption in optimization, needed for establishing convergence rates of gradient-based algorithms (see, for example, Section 1.2.2 in [1]). It is satisfied by a broad class of objective functions, including the least-squares data-fidelity term used in our experiments, as well as neural networks (e.g., Appendix B in [2]). Boundedness is a mild assumption, since it is always true for images that have bounded pixel values [0, 255].
>
> [1] Nesterov, Introductory Lectures on Convex Optimization, 2004
> [2] Hurault et al. Gradient step denoiser for convergent plug-and-play. ICLR, 2022.
>
> >3. *..., but no ablation study quantifies how ShaRP’s gains scale with ensemble size b.*
>
> Section E.2 of the supplementary material presents an ablation study quantifying performance gains with varying ensemble size (b). Computationally, ShaRP remains efficient because it uses only one restoration model from the ensemble at each inference step, thus having comparable computational complexity to single-model methods. The revised manuscript will ensure that this study is clearly referenced in the main manuscript for visibility.
>
> >4. *SISR uses only two Gaussian blurs, which is not enough.*
>
> Thanks for your suggestion. Prompted by your comment, we ran an additional SISR experiment using a new Gaussian kernel (σ = 1.0). As shown in the table below, we can achieve consistently good performance as the other two kernels in the manuscript. Note how ShaRP still provides excellent performance, suggesting that it would work with more Gaussian blurs.
>
> | Method    | PSNR  | SSIM  | LPIPS | FID   |
> |-----------|-------|-------|--------|--------|
> | DPIR      | 28.45 | 0.854 | 0.247  | 82.90  |
> | DDRM      | 27.26 | 0.803 | 0.209  | 44.77  |
> | DiffPIR   | 28.37 | 0.841 | 0.215  | 40.59  |
> | DRP       | 28.43 | 0.853 | 0.236  | 75.29  |
> | ShaRP     | 28.70 | 0.858 | 0.226  | 69.75  |
>
> >5. *Perceptual quality requires human evaluation.*
>
> Prompted by your comment, we introduced FID and LPIPS as perceptual metrics. Please refer to the rebuttal comment (7) to Reviewer DuT7 for the table.  The revised manuscript will mention that true perceptual quality requires human evaluation.
>
> >6. *The two papers shown below handle similar tasks: [1] Osmosis, ECCV, 2024. [2] DreamClean, ICLR, 2024.*
>
> We will cite these papers in the revised manuscript.
>
> >7 *Technical terms (e.g., "restoration operator") are inconsistently defined.*
>
> We will review the use of “restoration operator” across to make sure it is consistent. Restoration operator in our paper refers to an operator that computes MMSE solution of eq. (3).
>
> >8. *Supplement should provide experiments with other degradation types.*
>
> We will include results reported in our response to your Comment 1 to the supplementary material.
>
> >9. *How does ShaRP’s performance/complexity scale with ensemble size?*
>
> Please refer to our response to your Comment 2.
>
> >10. *Could ShaRP adapt to settings without even partially sampled data?*
>
> We are not entirely sure what the reviewer means by “without even partially sampled data.” If the reviewer is referring to the setting where there are no measurements at all, we can still run our method, but it was not conceived for this setting. Indeed, it would be very interesting to extend our methodology to “unconditional generation” or “sampling”, i.e., generation of images from the prior specified using a set of restoration operators. This will be a great direction for future work.
>
> >11 *How did inexact restoration operators impact convergence in practice? Does the bias term remain stable?*
>
> See Figure 7 in Section C.3, which shows stable convergence even with inexact (self-supervised) MMSE restoration operators. Overall, we did not observe any stability issues with ShaRP.

---

> > ### Comment · Reviewer_w5bu · 2025-04-05
> >
> > I appreciate the answers given and keep my score unchanged.

---

> > > ### Author Response · Authors · 2025-04-07
> > >
> > > Dear Reviewer w5bu,
> > > thanks for reading our rebuttal. Please let us know if there is anything we could do to make you consider increasing your rating.

---

### Official Review · Reviewer_DuT7 · 2025-02-24

**Overall Recommendation:** 4

**Summary:**

This paper introduces ShaRP, a stochastic regularization for linear inverse problems of the form y=Ax+e. This regulatization relies on MMSE (approximated) restoration machines R(*) for problems of the form s=Hx+n, where H is randomly chosen. By injecting the gradient of this regularization within the iterative recovery algorithm, and choosing at each itreration a differnt H to serve, ShaRP leads to improved recovery results in terms of PSNR, SSIM, (and LPIPS for SISR tests).

**Claims And Evidence:**

The claims made throughout the paper are well-supported.

**Essential References Not Discussed:**

A reference to RED-Diff is very much missing. See here: https://openreview.net/forum?id=1YO4EE3SPB

**Experimental Designs Or Analyses:**

Several comments above suggest that the comparisons are not complete and not fair:
- FID/KID results are missing - does ShaRP aim for high perceptual quality? If not, say so, but then, what is it aiming for?
- MMSE restoration is missing

**Methods And Evaluation Criteria:**

It is unclear what ShaRP is after - is it a better MAP estimate? or perhaps an MMSE one? Clearly, this is not a posterior sampler. This question is critical, because the tables in the results' section suggest that we are after an MMSE solution with the best PSNR, and the visual results (Figure 2 and 3) support this belief, as they they tend to be somewhat blurry.

In any case, a directly trained MMSE solver should be included in the comparisons, as it is nothing but a special case of the R operator that has been learned.

Also, as this work compares various different methods that strive for different goals (e.g. DPS and DDNM as posterior samplers), this must be accompanied with perceptual quality evaluation.

**Other Comments Or Suggestions:**

In equation 3, why not have sigma be random as well? In this case you get closer to RED-Diff mentioned above.

Equation 6 is very hard to follow, and esspecially so since you do not show the chosen h(*) beforehand. It would be helpfull to show how the actual h(*) is generalized to lead to (6).

**Other Strengths And Weaknesses:**

Strengths:
- Beautiful theoretical work
- Lovely idea that extends beyond RED and DRP.

Weaknesses:
- The paper is hard to follow in Section 3 - the order of the presentation is flawed, as it starts by referring to Algorithm 1, never define h(*), then jumps to talk about the gradient of f(*) - it is nearly impossible to follow.
- A clear definition of the obje3ctive or ShaRP is missing, and this could give better context to the results obtained.

**Questions For Authors:**

None.

**Relation To Broader Scientific Literature:**

OK in general

**Theoretical Claims:**

The theoretical part of this paper is solid and beautiful.

---

> ### Author Rebuttal · Authors · 2025-04-01
>
> We thank the reviewer for their valuable feedback.
>
> > 1. *It is unclear what ShaRP is after - is it a better MAP estimate? or perhaps an MMSE one? Clearly, this is not a posterior sampler. This question is critical, ..., as they they tend to be somewhat blurry.*
>
> This is a great point. ShaRP is not designed to produce classical MAP or MMSE estimates, nor is it a posterior sampler. Instead, it optimizes a novel objective function (Eq. (2) using h(x) in Eq. (6)) that balances data fidelity with an implicit regularization learned from an ensemble of priors. While the resulting reconstructions may exhibit characteristics similar to MMSE solutions (e.g., in terms of PSNR and visual smoothness), this is a consequence of the learned objective rather than an explicit design goal. We acknowledge that exploring sampling-based extensions could potentially enhance perceptual performance, and we will include this as a direction for future work.
>
> > 2. *In any case, a directly trained MMSE solver should be included ...*
>
> Thank you for the suggestion. For 4× CS-MRI, we added E2E-VarNet as a new baseline, which is a dedicated MMSE solver with data-fidelity constraint that achieves higher PSNR but is task-specific (i.e., [*] needs to be retrained for each task), effectively representing an upper bound on MSE performance.
>
> | Method        | PSNR (σ=0.05) | SSIM (σ=0.05) | PSNR (σ=0.10) | SSIM (σ=0.10) | PSNR (σ=0.15) | SSIM (σ=0.15) |
> |--------------|---------------|---------------|---------------|---------------|---------------|---------------|
> | Zero-filled  | 26.93         | 0.848         | 26.92         | 0.847         | 26.90         | 0.848         |
> | TV           | 31.17         | 0.923         | 31.08         | 0.921         | 30.91         | 0.915         |
> | PnP-FISTA    | 35.88         | 0.938         | 31.14         | 0.894         | 30.32         | 0.846         |
> | PnP-ADMM     | 35.76         | 0.941         | 32.36         | 0.878         | 30.66         | 0.838         |
> | DRP          | 35.52         | 0.936         | 32.32         | 0.914         | 30.57         | 0.901         |
> | DPS          | 32.62         | 0.888         | 31.39         | 0.870         | 30.29         | 0.856         |
> | DDS          | 35.21         | 0.937         | 35.03         | 0.935         | 34.51         | 0.925         |
> | ShaRP        | 37.59         | 0.963         | 35.81         | 0.951         | 34.92         | 0.942         |
> | E2E-VarNet [*] | 38.10       | 0.971         | 36.80         | 0.967         | 35.79         | 0.954         |
>
>
> > 3. *Also, as this work compares various different methods ... perceptual quality evaluation.*
>
> See our answer to Comment 4 below.
>
> > 4. *Several comments above suggest that the comparisons are not complete and not fair: FID/KID results are missing - does ShaRP aim for high perceptual quality? If not, say so, but then, what is it aiming for? MMSE restoration is missing.*
>
> We clarify that ShaRP primarily aims for high reconstruction accuracy rather than explicitly targeting perceptual quality. Nevertheless, we agree perceptual quality metrics provide valuable insights (in particular regarding the Perception Distortion tradeoff [3]). Hence, we have included FID alongside PSNR, SSIM, and LPIPS to offer a balanced assessment:
> | Method   | PSNR  | SSIM  | LPIPS | FID   |
> |----------|-------|-------|--------|--------|
> | DPIR     | 27.90 | 0.803 | 0.314  | 89.18  |
> | DPS      | 24.50 | 0.657 | 0.403  | 50.33  |
> | DiffPIR  | 28.59 | 0.834 | 0.172  | 46.12  |
> | DRP      | 28.24 | 0.836 | 0.235  | 64.23  |
> | ShaRP    | 29.28 | 0.872 | 0.209  | 58.79  |
>
> [3] Y. Blau., and T. Michaeli. "The perception-distortion tradeoff." CVPR, 2018.
>
> > 5. *A reference to RED-Diff is very much missing.*
>
> We will cite RED-Diff in the revised manuscript.
>
> > 6. *The paper is hard to follow in Section 3 - the order of the presentation is flawed ...*
>
> This is valuable feedback. We will implement the following changes: We will begin by clearly defining all necessary functions and terms, with a specific definition of h(). Following this, we will introduce Algorithm 1. We believe this revised structure will significantly improve the clarity and flow of Section 3.
>
>
> > 7. *A clear definition of the obje3ctive or ShaRP is missing ...*
>
> The objective function minimized by ShaRP is the composite function consisting of a data fidelity term g and a novel regularizer h specified in eq. (6). See also our response to Comment 1.
>
> > 8. *In equation 3, why not have sigma be random as well? ...*
>
> While ShaRP generates a random noise at each iteration, it does so using fixed sigma. Using random sigma is an interesting idea that we could explore in the future.
>
> > 9. *Equation 6 is very hard to follow, and esspecially so since you do not show the chosen h() beforehand. It would be helpfull to show how the actual h() is generalized to lead to (6).*
>
> We will revise the paper to better explain Equation (6).

---

> > ### Comment · Reviewer_DuT7 · 2025-04-03
> >
> > I appreciate the answers given nd change my grade to 4-accept.

---

> > > ### Author Response · Authors · 2025-04-04
> > >
> > > Thank you for reading our comments and raising the score. We will include your valuable feedback into the revised manuscript.

---

### Official Review · Reviewer_dZ9e · 2025-03-13

**Overall Recommendation:** 3

**Summary:**

The paper presents Stochastic Deep Restoration Priors (ShaRP), a framework for imaging inverse problems that leverages an ensemble of pre-trained deep restoration models. ShaRP uses a stochastic gradient descent approach where, in each iteration, a random degradation operator (with added Gaussian noise) is applied to simulate degradations. It minimizes a composite objective combining a data fidelity term with a regularizer derived from MMSE restoration operator score functions.

**Claims And Evidence:**

While the paper presents theoretical derivations and experimental results to support its contributions, some claims are not fully substantiated by clear and convincing evidence. For example, the claim that ShaRP can robustly handle diverse inverse problems without retraining—especially under self-supervised conditions from incomplete measurements—is supported by only a limited set of experiments and comparisons. Additionally, the theoretical guarantees regarding convergence and the robustness of the proposed regularizer would benefit from more extensive empirical validation and ablation studies.

**Essential References Not Discussed:**

While the paper cites a broad range of works on plug‐and‐play priors, denoising-based regularization, and diffusion models, it omits a few related works that could provide additional context for its contributions. For example, "Deep Image Prior" demonstrates that the inherent structure of an untrained convolutional network can serve as a powerful image prior, which is relevant to understanding unsupervised restoration strategies. Additionally, more recent self-supervised approaches such as Noise2Void and Self2Self that learn priors from corrupted or incomplete data could further clarify the paper’s context in scenarios without fully sampled measurements.

**Experimental Designs Or Analyses:**

The experimental design was examined for MRI reconstruction and single-image super-resolution tasks using standard benchmarks and evaluation metrics, with comparisons made against established baselines. However, there are several issues: details on dataset splits, noise levels, and degradation patterns are sometimes insufficient, which could affect reproducibility; baseline methods may not be equally tuned, potentially affecting fairness; and the analysis lacks extensive ablation and sensitivity studies to clearly disentangle the contributions of the ensemble approach and MMSE-based regularizer.

**Methods And Evaluation Criteria:**

The proposed methods and evaluation criteria appear to be appropriate for the targeted imaging inverse problems. The selection of tasks like MRI reconstruction and single image super-resolution provides a relevant and practical basis for evaluation.

**Other Comments Or Suggestions:**

A discussion on the practical computational cost of using an ensemble of restoration models, especially compared to single-model baselines, would be beneficial.

**Other Strengths And Weaknesses:**

The paper is original in its integration of multiple pre-trained restoration models to serve as image priors, moving beyond the traditional reliance on single Gaussian denoisers. This ensemble approach, coupled with a novel MMSE-based regularizer, is a contribution that addresses diverse inverse problems more flexibly. However, some weaknesses include idealized assumptions in the theoretical proofs and a need for more extensive ablation studies and sensitivity analyses to fully validate the empirical findings.

**Questions For Authors:**

Could you provide a more detailed description of the assumptions underlying your convergence proofs, particularly regarding the smoothness and Lipschitz continuity of the restoration operators?

Have you conducted ablation studies to isolate the impact of the ensemble approach versus the MMSE-based regularizer component?
What is the computational overhead of using an ensemble of restoration models compared to single-model approaches?

**Relation To Broader Scientific Literature:**

It builds upon the extensive literature on plug-and-play priors and regularization by denoising by moving beyond single Gaussian denoisers to an ensemble of restoration models, echoing ideas seen in recent works on deep restoration priors and stochastic denoising regularization. Additionally, it relates closely to diffusion model approaches that leverage score functions for sampling, while providing a novel MMSE-based regularizer that bridges the gap between traditional optimization methods and deep learning.

**Theoretical Claims:**

The proofs assume a level of smoothness and differentiability for the restoration operators that may not always hold for complex deep networks.
Certain Lipschitz continuity and boundedness conditions are assumed without full justification.
The convergence analysis is presented under idealized conditions, and the impact of real-world deviations from these assumptions is not fully explored.

---

> ### Author Rebuttal · Authors · 2025-04-01
>
> We thank the reviewer for their valuable feedback.
> > Response to concerns in Claims and Evidence
>
> 1. Generalization over configurations: We wanted to highlight that the supplementary material contains evidence supporting the robustness of ShaRP across diverse inverse problems without retraining, particularly under self-supervised conditions from incomplete measurements. Specifically:
> As detailed in Section C, we conducted experiments in the CS-MRI setting evaluating ShaRP's performance under a range of challenging conditions, including:  (1) Different undersampling rates (4x, 6x), (2) Varied mask types (Cartesian uniform, Cartesian random, and 2D Gaussian), (3) Multiple noise levels (0.005, 0.1, 0.15).
> These results demonstrate ShaRP's ability to handle diverse inverse problems without retraining, even with self-supervised learning from incomplete data.
>
> 2. Convergence Stability and Robustness: Section C.3 of the supplementary material presents an analysis of ShaRP's convergence stability and robustness. We evaluated this using both supervised and self-supervised MMSE restoration priors, providing empirical support for our theoretical guarantees.
>
> Prompted by the reviewer comment, the revised paper will include:
> (1) A summary table that consolidates the key ablation results across different undersampling rates, mask types, and noise levels. This will provide a clearer and more accessible overview.
> (2) Add a brief discussion in the main text highlighting these empirical findings and explicitly linking them to the claim of robustness without retraining.
> We believe these additions will strengthen our paper by highlighting extensive evidence provided in the paper.
>
> > Response to concerns in Theoretical Claims
>
> Prompted by the reviewer the revised paper will better highlight the following:
> 1. Lipschitz continuity and boundedness are needed only for Proposition 1. They are not needed for our main results—Theorem 1 and Theorem 2.
> 2. Lipschitz continuity is a standard and widely-used assumption in optimization for establishing convergence rates of gradient-based algorithms (see, for example, Section 1.2.2 in [1]). It is satisfied by a broad class of objective functions, including the least-squares data-fidelity term used in our experiments. The smoothness of MMSE restoration operators is well-known and has been extensively discussed in literature (see, for example, [2]).
> 3. Boundedness is a mild assumption, since it is always true for images that have bounded pixel values [0, 255].
> 4. Figure 7 in Section C.3 shows empirically convergence of our method in a real-world setting.
>
> [1] Nesterov, Introductory Lectures on Convex Optimization, 2004
>
> [2] Gribonval and Machart, Reconciling “priors” & “priors” without prejudice?, NeurIPS 2013.
>
> >  Response to concerns in Experimental Designs Or Analyses
>
> 1. Prompted by your comment, we will include a table summarizing all the relevant information about the experiment to guarantee reproducible results. We will additionally release our code upon acceptance, which should further simplify reproducibility.
>
> 2. Section E.2 of the supplementary material shows clear performance gains as ensemble size increases.
>
> > Respond to concerns in Essential References Not Discussed
>
> Thank you for pointing to  relevant works such as "Deep Image Prior," "Noise2Void," and "Self2Self." We will cite and discuss these works in our revised manuscript.
>
> > Response to concerns in Other Strengths And Weaknesses
>
> We appreciate the reviewer's assessment of our paper's originality and contributions. Following your recommendations, the revised manuscript will include summary tables pointing to the relevant results and will improve the discussion on the assumptions (see response to your Comment 2).
>
> > Response to concerns in Other Comments Or Suggestions
>
> The revised manuscript will explicitly mention that the computational cost of running ShaRP is comparable to those of single-model approaches. This is due to the stochastic nature of our algorithm that uses a $single$ restoration operator in each iteration.
>
> > Response to concerns For Authors
>
> 1. The smoothness of h associated with the MMSE restoration operators is well-known (see for example [1] or Appendix B of [2]). The smoothness can also be seen in the derivation presented in Appendix A of our paper. This is a direct consequence of the smoothness of the p(s | H) in eq. (5), since it is a Gaussian convolved with the prior p(x). Note also that since our restoration operator is implemented as a neural network with smooth activation functions like eLU, it will be inherently Lipschitz continuous [2].
> 2. (1) The detailed results of ablation studies isolating the effect of the ensemble are available in Section E.2, which is in the supplementary material. (2) There is no computational overhead due to our approach since only one model from the ensemble is utilized for each restoration instance. We will explicitly state this in the revised manuscript.

---

> > ### Comment · Reviewer_dZ9e · 2025-04-08
> >
> > Based on the authors' substantive response and alignment with peer reviews, I upgrade my score to Accept.

---

> > > ### Author Response · Authors · 2025-04-09
> > >
> > > Thank you for reading our comments and raising the score; we will include your valuable feedback to improve the manuscript.

---

### Official Review · Reviewer_X9oP · 2025-03-14

**Overall Recommendation:** 5

**Summary:**

The paper develops an plug-and-play imaging restoration algorithm that can use MMSE estimators trained to solve an arbitrary inverse problem involving linear forward operators and white gaussian noise (not just denoising), to solve a target linear inverse problem. That is, one can, for example, use a network trained to deblur and denoise images to solve a compressive sensing restoration task. The authors show the algorithm corresponds to taking biased stochastic gradients wrt the true MMSE loss. The proposed method is tested on compressive MRI and and single image super-resolution and is shown to generally outperform existing methods.

**Claims And Evidence:**

Claims are supported with evidence.

**Essential References Not Discussed:**

May be worth discussing the following paper:
Bansal, Arpit, Eitan Borgnia, Hong-Min Chu, Jie Li, Hamid Kazemi, Furong Huang, Micah Goldblum, Jonas Geiping, and Tom Goldstein. "Cold diffusion: Inverting arbitrary image transforms without noise." Advances in Neural Information Processing Systems 36 (2023): 41259-41282.

**Experimental Designs Or Analyses:**

Evaluation criteria are appropriate

**Methods And Evaluation Criteria:**

The methods and eval make sense

**Other Comments Or Suggestions:**

It might be better to rewrite (1) and (2) with complex-valued forward models, given the focus on MRI.

At least in the context of an ensemble of blur-then-deblur operations, ShaRP can be interpretted as masking then restoring a portion of k-space: ShaRP could be viewed as an ensemble of masked (denoising) autoencoders.

A subcript A in $\nabla g(x)$ in Algorithm 1 (i.e., $\nabla g_A(x)$) might be useful to remind readers where the forward model comes into play in the reconstruction algorithm. (Or just some additional annotation that g(x)) is the data fidelity term.)

What is s' in (12)? Are the losses ell_sup and ell_self wrt \bar{x}? The notation in this section could use more annotation.

**Other Strengths And Weaknesses:**

This is a well-written paper that studies an important problem. The proposed method is reasonably novel and apparently effective.

**Questions For Authors:**

In the MRI restoration task, the trained restoration algorithms are solving problems very similar to the target application; both are solving subsampled MRI, just with different masks. Is the proposed method still effective when the restoration algorithms are solving a very different task? E.g., can I solve compressive MRI by regularizing with an algorithm trained to perform image deblurring or inpainting?

**Relation To Broader Scientific Literature:**

The paper presents a thorough overview of the existing literature.

The paper differentiates itself from related self-supervised methods by stating "Ambient DMs seek to sample from px using DMs trained directly on undersampled measurements. Thus, during inference Ambient DMs assume access to the image prior px, while ShaRP only assumes access to the ensemble of likelihoods of multiple degraded observations."
However, because p_x was learned directly from undersampled measurements, the assumptions on Ambient GAN and related methods don't seem any more restrictive than those on the current method (which I would argue has also implicitly learned p_x).

"We introduce a novel regularization concept for inverse problems that encourages solutions that produce degraded versions closely resembling real degraded images." This seems conceptually similar to the equivariant imaging concept. Can the authors comment on the relationship between the two works. Is there a relationship between Theorem 2 and equivariant imaging?

**Theoretical Claims:**

The theory seems correct, though Section 4.3 could use further explanation.

---

> ### Author Rebuttal · Authors · 2025-04-01
>
> We thank the reviewer for feedback and thoughtful comments on our work.
>
> > 1. *The paper differentiates itself from related self-supervised methods by stating "Ambient DMs seek to sample from px using DMs trained directly on undersampled measurements. Thus, during inference Ambient DMs assume access to the image prior px, while ShaRP only assumes access to the ensemble of likelihoods of multiple degraded observations." However, because p_x was learned directly from undersampled measurements, the assumptions on Ambient GAN and related methods don't seem any more restrictive than those on the current method (which I would argue has also implicitly learned p_x).*
>
> Indeed, both Ambient DMs and ShaRP indeed implicitly learn about p(x) and we will revise that section of the paper to make this point clear.
>
> > 2. *We introduce a novel regularization concept for inverse problems that encourages solutions that produce degraded versions closely resembling real degraded images." This seems conceptually similar to the equivariant imaging concept. Can the authors comment on the relationship between the two works. Is there a relationship between Theorem 2 and equivariant imaging?*
>
> This is a very interesting perspective. Our approach can indeed be seen as finding a fixed point that exhibits equivariance under multiple degradations. The revised paper will include this nice interpretation suggested by the reviewer.
>
> > 3. *May be worth discussing the following paper: Bansal, Arpit, Eitan Borgnia, Hong-Min Chu, Jie Li, Hamid Kazemi, Furong Huang, Micah Goldblum, Jonas Geiping, and Tom Goldstein. "Cold diffusion: Inverting arbitrary image transforms without noise." Advances in Neural Information Processing Systems 36 (2023): 41259-41282.*
>
> We will cite and discuss Cold Diffusion in the revised version. Both Cold Diffusion and Ambient Diffusion (mentioned above) point toward a promising direction for extending our method to sampling, which we will highlight as a potential avenue for future work.
>
> > 4. *It might be better to rewrite (1) and (2) with complex-valued forward models, given the focus on MRI.*
>
> We will fix this per your suggestion.
>
> > 5. *At least in the context of an ensemble of blur-then-deblur operations, ShaRP can be interpreted as masking then restoring a portion of k-space: ShaRP could be viewed as an ensemble of masked (denoising) autoencoders.*
>
> Indeed, when considering an ensemble of blur-then-deblur operations, ShaRP can be seen as masking and subsequently restoring portions of k-space—akin to the mechanism of masked (denoising) autoencoders. In this interpretation, MMSE restoration networks effectively reconstruct the missing null-space signals. We will incorporate this connection into the discussion section of our revised manuscript.
>
> > 6. *A subscript A in $\nabla(g)$ in Algorithm 1 (i.e., $\nabla_A(g)$ ) might be useful to remind readers where the forward model comes into play in the reconstruction algorithm. (Or just some additional annotation that g(x)) is the data fidelity term.)*
>
> We will modify the notation accordingly.
>
> > 7. *What is s' in (12)? Are the losses ell_sup and ell_self wrt \bar{x}? The notation in this section could use more annotation.*
>
> In Eq. (12),  s’ refers to an independently subsampled measurement, defined as  s’ = P’M with P’ denoting a separate sampling pattern. We will revise the notation and add clarifying annotations in the revised version.
>
> > 8. *In the MRI restoration task, the trained restoration algorithms are solving problems very similar to the target application; both are solving subsampled MRI, just with different masks. Is the proposed method still effective when the restoration algorithms are solving a very different task? E.g., can I solve compressive MRI by regularizing with an algorithm trained to perform image deblurring or inpainting?*
>
> Section E.1 of our paper explores the scenario suggested by the reviewer by applying a pre-trained super-resolution (SR) model to the compressive sensing MRI problem. Despite the task mismatch, the SR prior still outperformed the Gaussian denoiser prior. We will edit the manuscript to ensure that it is clear this experiment was included.

---

### Decision · Program_Chairs · 2025-05-01

**Decision:**

Accept (poster)

**Comment:**

The reviewers provided a detailed assessment of the paper and found it to provide good contributions to the domain of inverse problems in imaging in terms of originality and theoretical analysis. There are some minor concerns on the experiments and several recent related methods have not been cited. The authors provided rebuttals and engaged in discussions with the reviewers. The reviewers unanimously concluded that the paper can be accepted at the conference and the AC agrees.